

# Combined retrieval of Arctic liquid water cloud and surface snow properties using airborne spectral solar remote sensing

André Ehrlich[1], Eike Bierwirth[1,2], Larysa Istomina[3], and Manfred Wendisch[1]

[1]Leipzig Institute for Meteorology (LIM), University of Leipzig, Leipzig, Germany
[2]now at: PIER-ELECTRONIC GmbH, Nassaustr. 33-35, 65719 Hofheim-Wallau, Germany
[3]Institute of Environmental Physics, University of Bremen, Bremen, Germany

*Correspondence to:* André Ehrlich (a.ehrlich@uni-leipzig.de)

**Abstract.** In the Arctic, the passive solar remote sensing of cloud properties over highly reflecting ground is challenging due to the low contrast between the clouds and underlying surfaces (sea ice and snow). Uncertainties in retrieved cloud optical thickness $\tau$ and cloud droplet effective radius $r_{\mathrm{eff,C}}$ may arise from uncertainties in the assumed spectral surface albedo, which is mainly determined by the commonly unknown snow effective grain size $r_{\mathrm{eff,S}}$. Therefore, in a first step this snow grain size effect is quantified systematically for a conventional bi-spectral retrieval of $\tau$ and $r_{\mathrm{eff,C}}$ for liquid water clouds. The largest impact of $r_{\mathrm{eff,S}}$ of up to 83 % on $\tau$ and 62 % on $r_{\mathrm{eff,C}}$ was found in case of small $r_{\mathrm{eff,S}}$ and optically thin clouds.

In the second part of the paper a retrieval method is presented that simultaneously retrieves all three parameters ($\tau$, $r_{\mathrm{eff,C}}$, $r_{\mathrm{eff,S}}$) in order to account for changes of the snow grain size in the cloud retrieval algorithm. Spectral cloud reflectivities at the three wavelength $\lambda_1 = 1040\,\mathrm{nm}$ (sensitive to $r_{\mathrm{eff,S}}$), $\lambda_2 = 1650\,\mathrm{nm}$ (sensitive to $\tau$), and $\lambda_3 = 2100\,\mathrm{nm}$ (sensitive to $r_{\mathrm{eff,C}}$) were normalized to reflectivity ratios and combined in a tri-spectral retrieval algorithm. Measurements collected by the Spectral Modular Airborne Radiation measurement sysTem (SMART-Albedometer) during the research campaign Vertical Distribution of Ice in Arctic Mixed-Phase Clouds (VERDI, April/May 2012) were used to test the retrieval procedure. Two cases of observations above the Canadian Beaufort Sea, one with dense snow-covered sea ice and another with a distinct sea ice edge were analyzed. The retrieved values of $\tau$, $r_{\mathrm{eff,C}}$, and $r_{\mathrm{eff,S}}$ consistently represented the cloud properties across this transition from snow-covered sea ice to the open water and were comparable to estimates based on satellite data. Analysis showed, that the uncertainties of the tri-spectral retrieval increase for high $\tau$, and low $r_{\mathrm{eff,S}}$, but nevertheless allows a simultaneous retrieval of cloud and surface snow properties providing snow effective grain size estimates in cloud-covered areas.

## 1 Introduction

During boreal winter, 15 % of the Earth's surface is covered by snow and sea ice (Vaughan et al., 2013), while clouds cover roughly two thirds of the globe (Boucher et al., 2013). Both, snow and clouds considerably increase the top of atmosphere albedo and, therefore, are eminent for determining the Earth's radiative energy budget. Changes in ice, snow and cloud cover indicate climate warming especially in Arctic areas (Wendisch et al., 2017). However, while snow and sea ice show a large seasonal variation, clouds may vary on shorter time and smaller spatial scales. Therefore, continuous observations of cloud and snow properties are important. Commonly, satellite remote sensing is used to monitor the most relevant radiative properties of





clouds and snow, such as cloud optical thickness and cloud droplet effective radius, snow cover and snow effective grain size and soot concentration (e.g., Stephens and Kummerow, 2007; Platnick et al., 2017; Zege et al., 2011; Painter et al., 2009).

In polar regions clouds cover large areas of snow, glaciers and sea ice, which complicates the retrieval of snow and cloud properties because the contrast between these bright surfaces and the clouds is low. At wavelengths typically used to retrieve

cloud properties from solar spectral reflectance measurements, the optical properties of clouds and snow are similar. Therefore, cloud retrieval algorithms utilize observations at wavelengths larger than 1000 nm where the snow albedo becomes lower. Measurements at 1500 nm wavelength, where snow strongly absorbs solar radiation, were used by Krijger et al. (2011) and Gao et al. (1998) to improve the cloud detection above snow surfaces. Furthermore, cloud optical thickness and droplet effective radius were successfully retrieved above snow surfaces by changing the channel combination of the retrieval algorithm of the

Moderate Resolution Imaging Spectroradiometer (MODIS) to the 1640 nm and 2130 nm band combination (Platnick, 2001; King et al., 2004).

However, the cloud reflectivity in this spectral range is affected by changes of the spectral albedo of the snow surface (Wiscombe and Warren, 1980). The smaller the snow grains, the higher the surface albedo becomes and the more radiation is reflected by the surface. Therefore, retrieving cloud properties (optical thickness and droplet effective radius) over snow

surfaces requires a precise assumption on the snow effective grain size below the clouds. Snow grain size may vary temporally and spatially due to new precipitation that reduces the snow grain size and because of snow metamorphism that slowly increases the snow grain size (e.g., Flanner and Zender, 2006; Jacobi et al., 2010). In polar areas, the snow effective grain size typically ranges between 50 μm for freshly fallen snow and 1000 μm for aged snow (Wiebe et al., 2013). This snow metamorphism changes the broadband surface albedo by 14 % from 0.89 to 0.77. The majority of this change occurs at longer wavelengths

where the imaginary part of the refractive index of ice is high. Therefor, the decrease of the spectral albedo at 1300 nm is enhanced to 65 % from 0.75 to 0.26 (Dang et al., 2016). In mid-latitude areas the snow metamorphism is often accelerated due to higher temperatures and may lead to snow effective grain sizes up to 3000 μm and an even stronger reduction of snow albedo (Singh, 2001; Derksen et al., 2014). Additionally, the albedo of white sea ice that is not covered by snow is reduced compared to snow-covered sea ice and, therefore, can be characterized by larger snow effective grain sizes (Malinka et al.,

2016). However, most cloud retrievals do not consider such a variation of the snow and sea ice albedo. For the MODIS cloud product Collection 6 only a fixed surface albedo of 0.03 for both wavelength bands 1640 nm and 2130 nm is assumed over sea ice or snow-covered areas (King et al., 2004). Different to land surfaces no spatial or temporal changes of the snow and sea ice albedo are considered.

Uncertainties of the surface albedo can significantly affect the retrieval of cloud optical properties (e.g., Fricke et al., 2014;

Rolland and Liou, 2001; Platnick, 2001). Most of these studies focus on optically thin clouds over typical land surfaces with high variability of the spectral albedo for wavelengths below 1 μm. For a thin cirrus, Fricke et al. (2014) estimated retrieval uncertainties of up to 50 % for the ice crystal effective radius depending on cloud optical thickness. Rolland and Liou (2001) showed that the retrieval uncertainties of thin cirrus can be improved by 20 % for optical thickness and by 45 % for ice crystal effective radius when an improved estimate of the surface albedo variability is applied. For snow-covered areas only the



difference between sea ice and ice free albedo has been addressed and considered in improved retrieval algorithms (Platnick, 2001). An estimate of how changes of the snow albedo effect cloud retrieval is missing in the literature so far.

For satellite observations of spectral solar radiation, retrieval algorithms that provide snow effective grain size have been developed by Zege et al. (2011); Painter et al. (2009). These techniques utilize the dependence of spectral snow albedo/reflectivity on the snow effective grain size. A larger snow effective grain size increases the photon path length within the snow grain and the probability that radiation is absorbed in the snow layer which would reduces the snow albedo. As already thin snow layers determine the radiation reflection by the surface, the retrievals are most sensitive to the uppermost snow layer and cover changes by precipitation and snow metamorphism (Wiebe et al., 2013; Libois et al., 2013). In addition, the retrieval algorithms by Zege et al. (2011) and Painter et al. (2009) estimate the black carbon concentration in the snow surface that mostly effects the visible range of the spectral albedo. Unfortunately, these satellite retrievals of snow properties may not cover the full spatial and temporal evolution of snow effective grain size and snow albedo as they are limited to cloud free areas (Lyapustin et al., 2009; Zege et al., 2011). However, especially in polar regions cloud layers that prevail for several days are frequently observed (Herman and Goody, 1976; Shupe et al., 2006, 2011). In that case the cloud retrieval may suffer from an outdated assumption on snow or sea ice albedo. A correct solution is only possible, if snow and cloud properties are determined in combination.

Measurements of spectral cloud reflectivity have been successfully applied to distinguish between liquid and ice water clouds (Pilewskie and Twomey, 1987; Ehrlich et al., 2008; LeBlanc et al., 2015). Making use of differences in the spectral absorption of liquid water and ice, the spectral shape of the cloud reflectivity can be analyzed to determine different indices to identify the dominant cloud phase. Similarly, this study makes use of the different spectral absorption characteristics for snow surfaces and liquid water clouds to develop a method that allows to retrieve cloud and snow properties simultaneous. To illustrate the need of such retrieval methods, in Section 2 the uncertainties due to uncertainties in the assumed snow albedo on the retrieval of cloud properties will be quantified. The new retrieval method will be introduced in Sections 3 and 4 including the identification of suited wavelengths applied in the retrieval and the forward simulations of cloud reflectivity that build the backbone of the retrieval. In Section 5, the algorithm that is limited to cases of liquid water clouds is applied to two specific cases which have been observed by airborne spectral cloud reflectivity measurements during the field campaign Vertical Distribution of Ice in Arctic Clouds (VERDI) over the Canadian Beaufort Sea in 2012.

## 2 Uncertainties of bi-spectral cloud retrieval over snow

### 2.1 Forward simulations

Based on radiative transfer simulations the impact of uncertainties of the snow albedo on the retrieved cloud properties is quantified. The spectral solar radiation above a liquid water cloud layer was simulated using the DISORT 2 radiative transfer solver embedded in the library for radiative transfer (libRadtran, Emde et al., 2016). A solar zenith angle of 63° representative for Arctic conditions around spring was chosen in the simulations. Typical Arctic boundary layer liquid water clouds located between 200 m and 500 m altitude were simulated. Cloud optical thickness $\tau$ was varied from 1 to 20; cloud droplet effective radius $r_{\mathrm{eff,C}}$ between 2 μm and 25 μm. For all clouds, simulations with different surface albedo covering snow effective grain





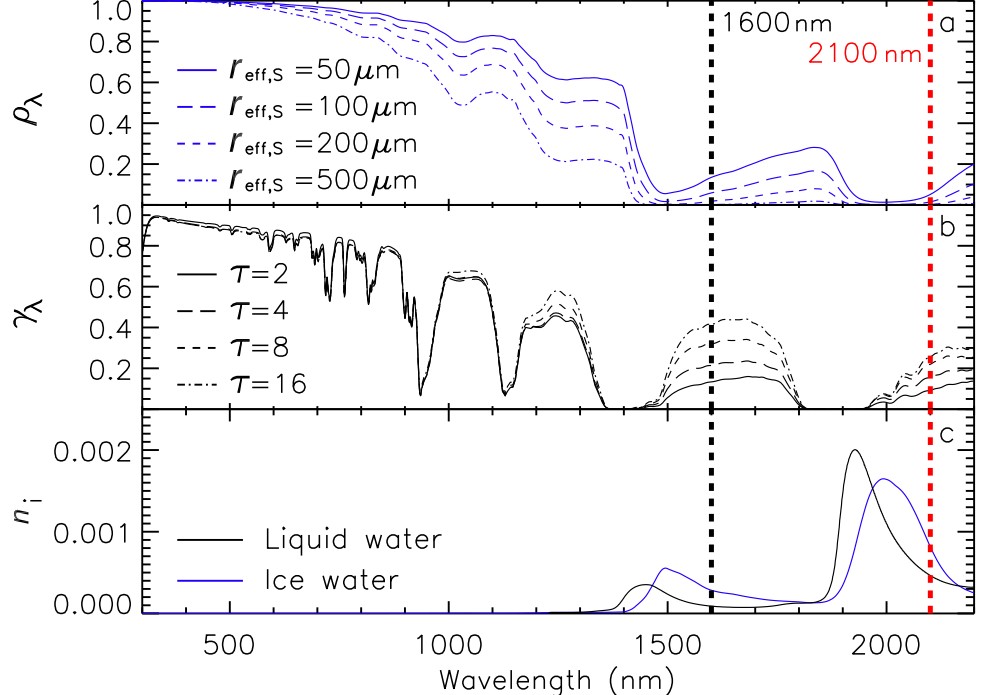

**Figure 1.** Comparison of spectral snow albedo $\rho_\lambda$ (a) and cloud reflectivity $\gamma_\lambda$ (b) for different $r_{\mathrm{eff,S}}$ and $\tau$, respectively. Clouds reflectivity has been simulated for clouds with $r_{\mathrm{eff,C}} = 10\,\mu\mathrm{m}$ that are located above a snow surface with $r_{\mathrm{eff,S}} = 100\,\mu\mathrm{m}$. The imaginary part of the refractive index of ice and liquid water is given in panel (c). Vertical lines indicate the wavelength used in the bi-spectral cloud retrieval.

sizes $r_{\mathrm{eff,S}}$ between $10\,\mu\mathrm{m}$ and $800\,\mu\mathrm{m}$ were performed. The spectral snow albedo $\rho_\lambda$ was calculated with the parametrization by Zege et al. (2011) using the refractive index of ice presented by Warren and Brandt (2008) and a form-factor $A = 5.8$. This form factor is adopted from Zege et al. (2011) and accounts for the non-sphericity of snow grains; it represents a mixture of randomly oriented hexagonal plates and columns with rough surfaces. This mixture and, therefore, the form-factor $A$ may

5      differ in reality depending on the local snow properties. However, an uncertainty of $A$ can be attributed to an uncertainty in the effective snow grain size as both properties have the same spectral impact on the snow reflection characteristics, such as spectral albedo. Snow impurities by black carbon were neglected as the absorption by black carbon is typically limited to wavelengths less than $1000\,\mathrm{nm}$ (e.g., Warren and Wiscombe, 1980; Liou et al., 2014) that are not used in cloud retrieval over snow surfaces. A set of calculated spectral snow albedo $\rho_\lambda$ is presented in Figure 1a which illustrates the decrease of $\rho_\lambda$

10      with increasing values of $r_{\mathrm{eff,S}}$ for wavelengths $\lambda > 1000\,\mathrm{nm}$ where the imaginary part of the refractive index of ice is high (Figure 1c).





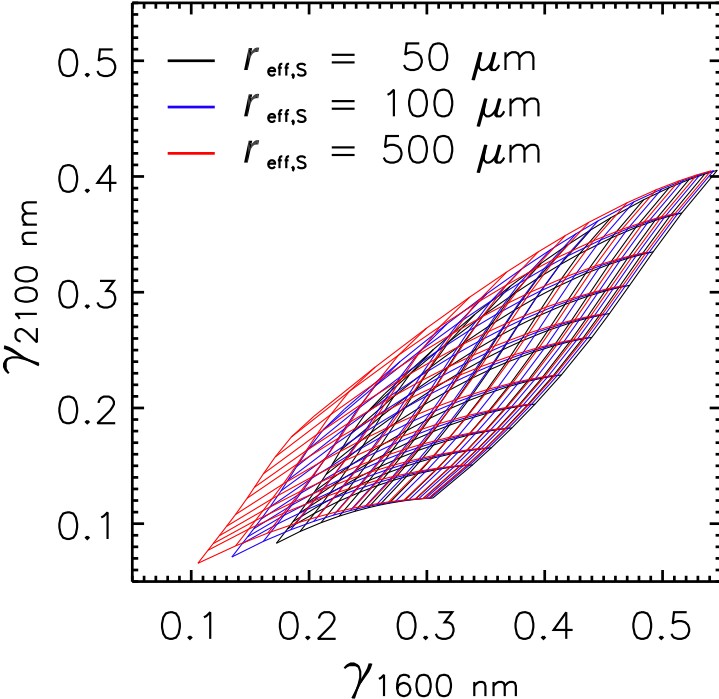

**Figure 2.** Bi-spectral retrieval grids of cloud top nadir reflectivity $\gamma_{1600\,nm}$ and $\gamma_{2100\,nm}$ assuming three different snow effective grain sizes $r_{\mathrm{eff,S}}$ of 50, 100, and 500 $\mu$m. The simulated reflectivities cover cloud optical thickness $\tau$ between 3 and 20 and cloud droplet effective radius $r_{\mathrm{eff,C}}$ between 6 $\mu$m and 25 $\mu$m.

The simulated upward nadir radiance $I_\lambda$ and downward irradiance $F_\lambda$ were converted into spectral cloud reflectivity $\gamma_\lambda$ defined by,

$$\gamma_\lambda = \frac{\pi \cdot I_\lambda}{F_\lambda}. \tag{1}$$

In Figure 1b a set of $\gamma_\lambda$ for typical values of $\tau$ between 2 and 16 is shown for a fixed cloud droplet size of $r_{\mathrm{eff,C}} = 10\,\mu$m
5  and a typical snow effective grain size of $r_{\mathrm{eff,S}} = 100\,\mu$m. The simulations illustrate that $\tau$ impacts $\gamma_\lambda$ only at wavelengths $\lambda > 1000\,$nm where the snow albedo is lower than 0.8.

Based on the simulated cloud reflectivities, which are used as synthetic measurements, a bi-spectral cloud retrieval was applied to obtain cloud optical thickness and droplet effective radius. This retrieval method is similar to the cloud product of MODIS (Platnick, 2001). The retrieval uses the dependence of $\gamma_{1600\,nm}$ (less-absorbing wavelength) to cloud optical thickness
10  and the dependence of $\gamma_{2100\,nm}$ (high-absorbing wavelength) to cloud droplet effective radius and follows the method by Nakajima and King (1990).



The bi-spectral retrieval grid obtained from the simulated cloud reflectivities is presented in Figure 2 for three snow effective grain sizes: 50, 100, and 500 $\mu$m. The grids significantly differ and show a snow grain size effect especially for low values of $\gamma_{1600\,nm}$, while at higher reflectivities the grids tend to converge. The reflectivity $\gamma_{2100\,nm}$ is less affected by changes of $r_{eff,S}$ as the snow albedo is close to zero for $r_{eff,S} > 100\,\mu$m (see Figure 1). This behavior suggests, that the retrieval of $\tau$, that is strongly linked to $\gamma_{1600\,nm}$, is primarily affected by the snow albedo. However, the non-rectangular shape of the grids indicate that also the retrieved $r_{eff,C}$ will be affected by a changing snow effective grain size as the reflectivities at both wavelengths are coupled to both cloud parameters.

## 2.2 Snow grain size effect on cloud retrieval results

To quantify the snow grain size effect on uncertainties of liquid water cloud retrieval results obtained over snow surfaces with unknown grain size, a set of retrieval assuming different values of snow effective grain sizes were performed for each synthetic measurement defined by $\tau$, $r_{eff,C}$, and $r_{eff,S}$. The purpose of this exercise is to use the synthetic measurement (for which the original snow effective grain size is known), and pretend not to know $r_{eff,S}$ when starting the retrieval. In the retrieval forward simulations different $r_{eff,S}$ are assumed to quantify the impact of this wrong assumption on the retrieved cloud properties. In Figures 3a and 3b the retrieval results are compared to the original cloud properties for synthetic measurements calculated with an original snow effective grain size of 50 $\mu$m, but retrieved from forward simulations assuming a snow effective grain size of 200 $\mu$m (crosses). The asterisks symbols in Figures 3a and 3b indicate the opposite case: originally $r_{eff,S} = 200\,\mu$m is used to produce the synthetic measurement and then $r_{eff,S} = 50\,\mu$m is assumed in the retrieval of the cloud properties. While Figure 3a shows retrieved $\tau$ for different $r_{eff,C}$ indicated by the color code, Figure 3b presents retrieved $r_{eff,C}$ for clouds of different $\tau$ also indicated by a color code.

Assuming $r_{eff,S}$ to be larger than originally present, the retrieved $\tau$ is systematically overestimated because the surface albedo is underestimated in this case (larger $r_{eff,S}$ assumes lower snow albedo at 1600 nm, see also Figure 1). If $r_{eff,S}$ is underestimated (surface albedo overestimated) the snow grain size effect is inverted leading to an underestimation of $\tau$. This is in agreement with the general surface albedo sensitivity of cloud retrieval as discussed by Rolland and Liou (2001) and Fricke et al. (2014). For the case presented in Figure 3, errors in the retrieval range up to 83 % for low optical thickness $\tau = 3$.

The results for $\tau$ do not significantly depend on $r_{eff,C}$. Contrarily, the uncertainties introduced in the retrieval of $r_{eff,C}$ strongly depend on $\tau$ as illustrated in Figure 3b. Especially for clouds of low optical thickness, the retrieved $r_{eff,C}$ is significantly overestimated/underestimated when the snow effective grain size is assumed to be higher/lower than originally present. The snow grain size effect ranges up to 62 % for optically thin clouds with small $r_{eff,C}$. In case larger snow effective grain sizes are assumed, the absorption observed in $\gamma_{1600\,nm}$ is overestimated while $\gamma_{2100\,nm}$ is almost unchanged. This combination leads to an estimate of too low $r_{eff,C}$ in the retrieval.

While the retrieval of $\tau$, independently of $r_{eff,C}$, is always biased due to uncertainties of $r_{eff,S}$, no snow grain size effect is observed for the retrieval of $r_{eff,C}$ in case $\tau$ is larger than about 10. For optically thick clouds, the high extinction of incoming radiation inside the cloud layer leads to a low amount of radiation that reaches the surface and interacts with the snow and





**Figure 3.** Comparison of synthetically retrieved $\tau$ (a, c) and $r_{\mathrm{eff,C}}$ (b, d) with the original parameter value. Calculations in panel a and b are performed for assuming a larger snow effective grain size of $r_{\mathrm{eff,S}} = 200\,\mu\mathrm{m}$ instead of the original $r_{\mathrm{eff,S}} = 50\,\mu\mathrm{m}$ (crosses) and a smaller snow effective grain size of $r_{\mathrm{eff,S}} = 50\,\mu\mathrm{m}$ instead of the original $r_{\mathrm{eff,S}} = 200\,\mu\mathrm{m}$ (asterisks). In panel c and d all combinations of assumed and original $r_{\mathrm{eff,S}}$ are analyzed for a specific cloud of $\tau = 4$ and $r_{\mathrm{eff,C}} = 10\,\mu\mathrm{m}$. The red dots in panel c and d indicate the cases included in panel a and b, where results for the same cloud are indicated by green circles.

is transmitted back through top of the cloud. In this case the interaction of radiation with the surface can be neglected and, therefore, the surface albedo, respectively the assumption of $r_{\mathrm{eff,S}}$, is not relevant.





The cases discussed in Figure 3a and 3b represent the typical range of $r_{\mathrm{eff,S}}$ from $50\,\mu m$ to $200\,\mu m$ as expected in Arctic areas for snow surfaces. However, white sea ice and snow cover in mid-latitudes may exhibit a higher variability leading to larger uncertainties. Therefore, Figures 3c and 3d summarize the snow grain size effect on the retrieval of $\tau$ (Figure 3c) and $r_{\mathrm{eff,C}}$ (Figure 3d) for a set of combinations of assumed and original $r_{\mathrm{eff,S}}$. The red dots indicate the cases included in Figure 3a

and 3b, where results for the same cloud are indicated by green circles. An exemplary cloud with low optical thickness of $\tau = 4$ and $r_{\mathrm{eff,C}} = 10\,\mu m$ was analyzed.

The over- and underestimation of $r_{\mathrm{eff,S}}$ leads to almost symmetric effects for the clouds investigated here. The maximum snow grain size effect on the retrieval of $\tau$ covered by the simulations leads to a retrieval of $\tau = 2$ or $\tau = 6$. For $r_{\mathrm{eff,C}}$ the results range between $6\,\mu m$ and $13\,\mu m$. The effects are most pronounced when either smaller snow effective grain sizes are assumed or

originally present; e.g., $50\,\mu m$ assumed but $300\,\mu m$ present, or $300\,\mu m$ assumed and $50\,\mu m$ present. Similar mismatch between assumed and original $r_{\mathrm{eff,S}}$ at larger grain sizes, e.g., $300\,\mu m$ vs. $500\,\mu m$, do cause much lower errors in the retrieved $\tau$ and $r_{\mathrm{eff,C}}$. This indicates, that especially in polar areas where snow grain sizes are typical smaller, the retrieval biases due to a wrong assumption of $r_{\mathrm{eff,S}}$ can not be neglected.

## 3   Separating the spectral signatures of liquid water clouds and snow

In case liquid water clouds are located above a snow surface, the spectral differences of absorption of solar radiation by snow (ice water) and clouds (liquid water) as illustrated in Figure 1 can be used to separate the surface and cloud contributions to the reflected radiation above the cloud. In addition, the different size ranges of cloud droplets (typically $r_{\mathrm{eff,C}} < 20\,\mu m$) and snow grains (typically $r_{\mathrm{eff,S}} > 50\,\mu m$) amplify these spectral signatures of ice and liquid water. Inside large snow grains the photon path length is prolonged, which leads to a stronger absorption and a lower reflectivity compared to the smaller liquid water

droplets.

Using the cloud reflectivity simulations introduced in Section 2, two measures to identify wavelengths that are most sensitive to only one single parameter, either $\tau$, $r_{\mathrm{eff,C}}$ or $r_{\mathrm{eff,S}}$, are derived. The first parameter $\sigma$ is provided by the mean standard deviation of $\gamma_\lambda$ with respect to a single parameter $\tau$, $r_{\mathrm{eff,C}}$, and $r_{\mathrm{eff,S}}$. The second parameter are the spectral weightings of a principle component analysis (PCA) applied to the full set of simulations. Corresponding to the cloud and snow parameters

changed in the simulations, the spectral weights $\Gamma_1$, $\Gamma_2$, and $\Gamma_3$ of the first three principle components are associated with $\tau$ ($\Gamma_1$), $r_{\mathrm{eff,C}}$ ($\Gamma_2$), and $r_{\mathrm{eff,S}}$ ($\Gamma_3$).

Both measures are shown in Figure 4 for $\tau$ ($\sigma_\tau$ and $\Gamma_1$), for $r_{\mathrm{eff,C}}$ ($\sigma_{r_{\mathrm{eff,C}}}$ and $\Gamma_2$), and for $r_{\mathrm{eff,S}}$ ($\sigma_{r_{\mathrm{eff,S}}}$ and $\Gamma_3$). The three calculated functions of $\sigma$ show that the maximum variability of the cloud reflectivity with respect to the three parameters is located in different spectral regions. While $r_{\mathrm{eff,S}}$ mostly affects the cloud reflectivity at wavelengths between $930\,nm$ and

$1350\,nm$, $\tau$ introduces high standard deviations in the wavelength range of $1500 - 1800\,nm$. However, this spectral range is also influenced by $r_{\mathrm{eff,C}}$ with $\sigma_{r_{\mathrm{eff,C}}}$ showing only moderately lower values than $\sigma_\tau$. Values of similar magnitude are observed for $\sigma_{r_{\mathrm{eff,C}}}$ at longer wavelength larger $2000\,nm$. In this spectral range $r_{\mathrm{eff,C}}$ is the dominating parameter determining the cloud reflectivity.





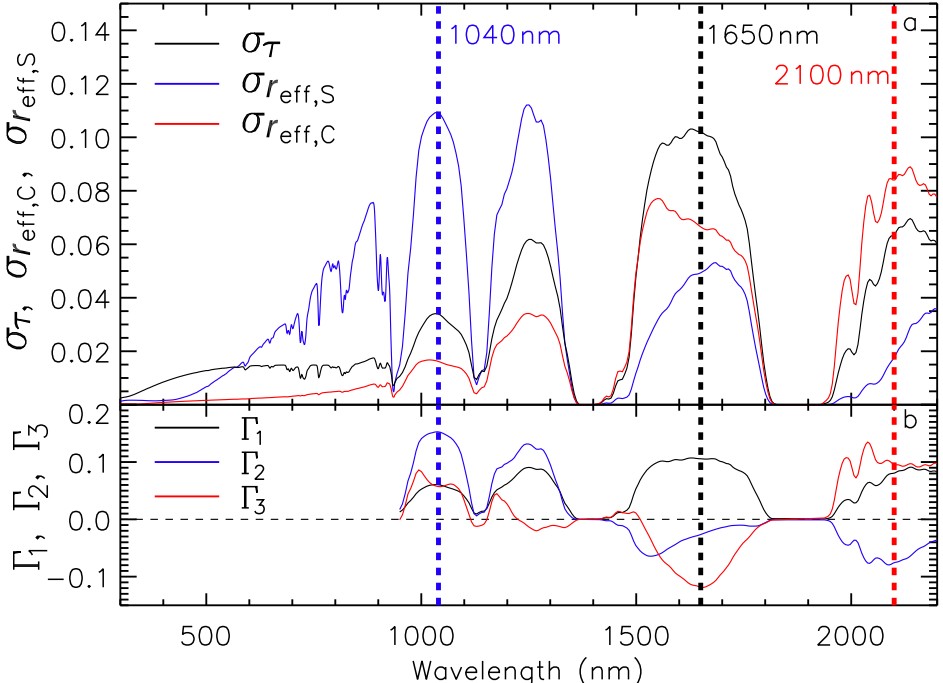

**Figure 4.** Mean standard deviations of spectral cloud reflectivity $\sigma_\tau$, $\sigma_{r_{\mathrm{eff,C}}}$, and $\sigma_{r_{\mathrm{eff,S}}}$ with respect to a single cloud or snow parameter $\tau$, $r_{\mathrm{eff,C}}$, and $r_{\mathrm{eff,S}}$ calculated for the sets of radiative transfer simulations (panel a). First three spectral weights $\Gamma_1$, $\Gamma_2$, and $\Gamma_3$ of a principle component analysis are given in panel b.

Similar spectral pattern result from the PCA, which delivers the spectral weighs $\Gamma_1$, $\Gamma_2$, and $\Gamma_3$ as presented in Figure 4b. A comparison with the three calculated $\sigma$ in Figure 4b reveals that $\Gamma_1$ can be associated with $\tau$, $\Gamma_3$ with $r_{\mathrm{eff,C}}$ and $\Gamma_2$ with $r_{\mathrm{eff,S}}$. The major contribution to $\Gamma_1$ is located in the wavelength range $930 - 1350\,\mathrm{nm}$ dominated by the changes in snow albedo, $r_{\mathrm{eff,S}}$. The impact of $\tau$ is spectrally neutral, showing a similar magnitude of $\Gamma_1$ in all analyzed wavelengths expect of the water vapor absorption bands. Contrarily, wavelengths above $1500\,\mathrm{nm}$ contribute most to the weight of the third principle component $\Gamma_3$, but with opposite signs for the $1500 - 1800\,\mathrm{nm}$ and the $2000 - 2200\,\mathrm{nm}$ wavelength range.

It has to be mentioned, that these sensitivities might change for different scenarios assumed in the radiative transfer simulations, e.g., different solar zenith angle, cloud altitude, profile of cloud droplet size, or aerosol concentration. However, the general separation of the three parameters by different spectral ranges will not essentially differ.





## 4 Tri-spectral retrieval algorithm

Based on the spectral signatures imprinted in the cloud reflectivity by variations of $\tau$, $r_{\mathrm{eff,C}}$ and $r_{\mathrm{eff,S}}$ a tri-spectral retrieval algorithm using measured $\gamma_\lambda$ is proposed to retrieve the three cloud and snow parameter simultaneously. Extending conventional bi-spectral cloud retrievals by a third measurement at a wavelength sensitive to $r_{\mathrm{eff,S}}$ adds the information on the snow

grain size. Compared to retrieval algorithms that rely on a fixed assumption of $r_{\mathrm{eff,S}}$ this approach can reduce the uncertainty of the retrieved cloud parameters.

Measurements at $\lambda_1 = 1040\,\mathrm{nm}$ most sensitive to $r_{\mathrm{eff,S}}$, $\lambda_2 = 1650\,\mathrm{nm}$ sensitive to $\tau$, and $\lambda_3 = 2100\,\mathrm{nm}$ sensitive to $r_{\mathrm{eff,C}}$ were chosen in the retrieval algorithm. To improve the separation of the individual cloud and snow parameters, the radiance-ratio method introduced by Werner et al. (2013); Brückner et al. (2014); LeBlanc et al. (2015) was applied. Here a normaliza-

tion with the cloud reflectivity at $\lambda_0 = 860\,\mathrm{nm}$ was chosen. The corresponding ratios $R_1$, $R_2$, and $R_3$ are calculated by,

$$R_1 = \frac{\gamma_{\lambda_1}}{\gamma_{\lambda_0}}, \qquad R_2 = \frac{\gamma_{\lambda_2}}{\gamma_{\lambda_1}}, \qquad R_3 = \frac{\gamma_{\lambda_3}}{\gamma_{\lambda_2}}, \tag{2}$$

$$\lambda_0 = 860\,\mathrm{nm}, \ \lambda_1 = 1040\,\mathrm{nm},$$

$$\lambda_2 = 1650\,\mathrm{nm}, \ \lambda_3 = 2100\,\mathrm{nm}.$$

These normalizations additionally reduce the uncertainties of the retrieval by cancelling potential biases in the radiometric calibration of the measurements. Alternatively, e.g., when restricted to MODIS channels, for $\lambda_2$ related to $r_{\mathrm{eff,S}}$ a wavelengths between $1200\,\mathrm{nm}$ and $1300\,\mathrm{nm}$ can be chosen where cloud reflectivity is still most sensitive to $r_{\mathrm{eff,S}}$.

Similar to Section 3, the mean standard deviation $\sigma$ with respect to a single parameter $\tau$, $r_{\mathrm{eff,C}}$, and $r_{\mathrm{eff,S}}$ was calculated

for the three reflectivity ratios. Table 1 shows $\sigma$ for all possible combinations. The higher $\sigma$, the more sensitive $R_1$, $R_2$, and $R_3$ are for changes of an individual cloud or snow parameter. For each parameter, $\tau$, $r_{\mathrm{eff,C}}$, and $r_{\mathrm{eff,S}}$, one reflectivity ratios shows a significant higher $\sigma$ compared to the other ratios. For the cloud optical thickness, the highest $\sigma = 0.145$ is found for $R_2$ and almost twice as high compared to the sensitivity of $R_1$ and $R_3$. Similarly, $R_1$ and $R_3$ show the highest sensitivity to $r_{\mathrm{eff,S}}$ and $r_{\mathrm{eff,C}}$, respectively, with $\sigma = 0.074$ and $\sigma = 0.121$. This indicates, that $R_1$, $R_2$, and $R_3$ are well suited to separate

the information of $\tau$, $r_{\mathrm{eff,C}}$, and $r_{\mathrm{eff,S}}$ from spectral reflectivity measurements.

**Table 1.** Standard deviation of the reflectivity ratios $R_1$, $R_2$, and $R_3$ with respect to one of the three cloud and snow parameter $\tau$, $r_{\mathrm{eff,C}}$, and $r_{\mathrm{eff,S}}$.

|  | $r_{\mathrm{eff,S}}$ | $\tau$ | $r_{\mathrm{eff,C}}$ |
|---|---|---|---|
| $R_1$ | **0.074** | 0.039 | 0.015 |
| $R_2$ | 0.065 | **0.145** | 0.088 |
| $R_3$ | 0.054 | 0.056 | **0.121** |



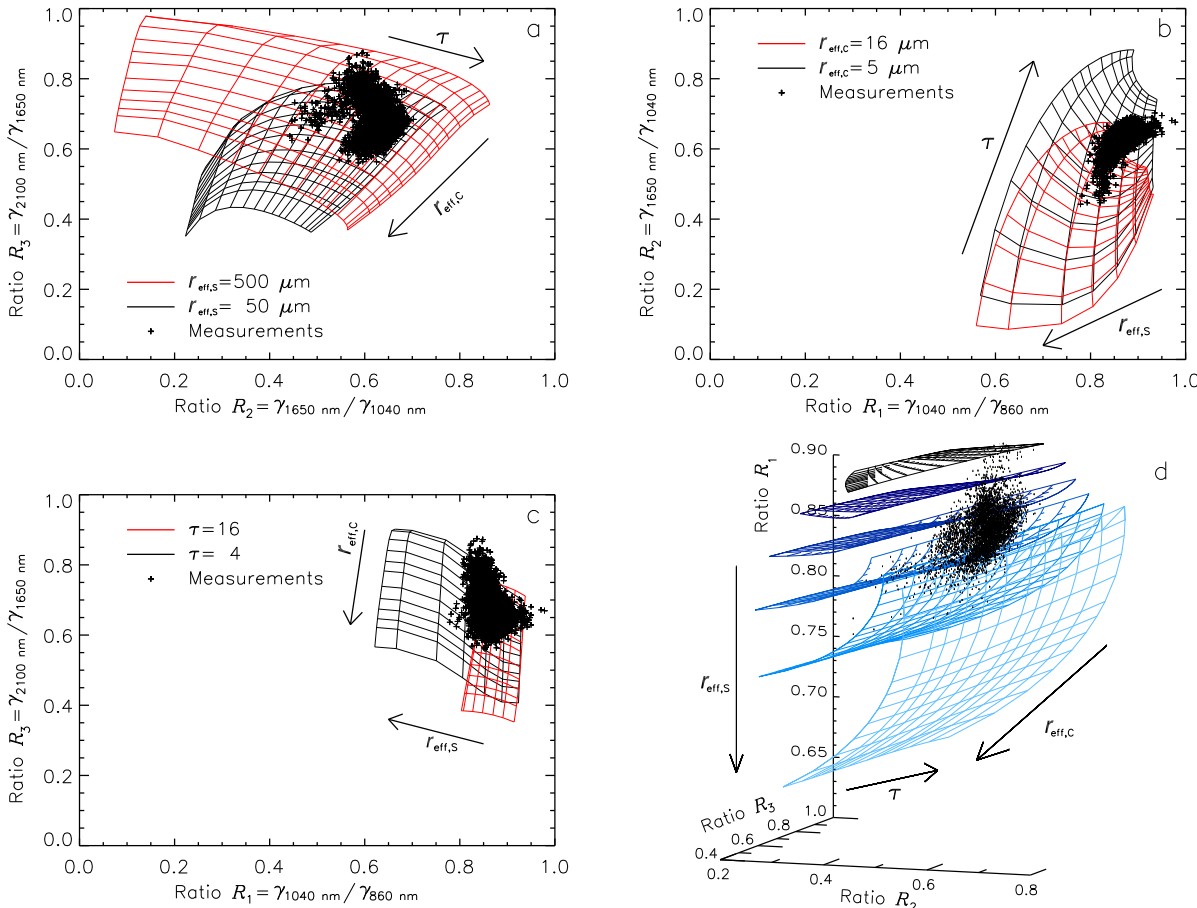

**Figure 5.** Three-dimensional retrieval grid using the ratios $R_1$, $R_2$, and $R_3$ obtained from simulations covering $\tau = 1\text{-}20$, $r_{\text{eff,C}} = 5\text{-}25\,\mu\text{m}$ and $r_{\text{eff,S}} = 25\text{-}800\,\mu\text{m}$. Panel a-c show each two selected two-dimensional section of the retrieval grid, while panel d covers the full 3D grid. Exemplary measurements obtained during VERDI (29 April 2012) are imprinted by black dots.

## 4.1 Retrieval grid

The set of cloud reflectivity simulations introduced in Section 2 served for the forward simulations (solar zenith angle of $63°$, liquid water cloud layer between $200\,\text{m}$ and $500\,\text{m}$). The simulated grids of $R_1$, $R_2$, and $R_3$ used for the tri-spectral retrieval are presented in Fig 5. While Fig. 5d shows the full 3-dimensional (3D) grid covering $\tau = 1\text{-}20$, $r_{\text{eff,C}} = 5\text{-}25\,\mu\text{m}$ and $r_{\text{eff,S}} = 25\text{-}800\,\mu\text{m}$, Fig. 5a-c present 2-dimensional projections of two reflectivity ratios with one parameter being fixed. In Fig. 5a, $r_{\text{eff,S}}$ is fixed to $50\,\mu\text{m}$ (black) and $500\,\mu\text{m}$ (red), while in Fig. 5b $r_{\text{eff,C}}$ is fixed to $5\,\mu\text{m}$ (black) and $16\,\mu\text{m}$ (red) and in Fig. 5c $\tau$ is fixed to a value of 4 (black) and 16 (red). In all figures the directions of change by $\tau$, $r_{\text{eff,C}}$, and $r_{\text{eff,S}}$ are indicated by arrows. The simulated grid shows that the individual lines cross almost perpendicular for large ranges of the simulated parameters.




No overlapping of the different surfaces is observed. This indicates that the three ratios $R_1$, $R_2$, and $R_3$ separate the influence of the three parameters $\tau$, $r_{\text{eff,C}}$, and $r_{\text{eff,S}}$ on the cloud reflectivity and allow a retrieval of the cloud and snow parameters leaving little ambiguities. Only for small snow effective grain sizes lower than $50\,\mu\text{m}$ the grid becomes more narrow as obvious in Fig. 5d (uppermost black grid) and Fig. 5a (black grid). Such a narrow grid increases the uncertainty of retrieved $\tau$ and

$r_{\text{eff,C}}$ especially for clouds of low optical thickness. This grid characteristic is caused by the higher surface albedo of snow for small $r_{\text{eff,S}}$ which reduces the contrast between cloud and snow surface also in the wavelength range used for the retrieval algorithm. However, a retrieval of cloud and snow properties in these ranges is still possible if the measurement uncertainties are sufficiently low.

    Ambiguities appear in the retrieval for small cloud droplet effective radii of $r_{\text{eff,C}} < 5\,\mu\text{m}$ (not shown in Fig. 5). In that case

the absorption of cloud droplets is to week and cloud reflectivity can be similar to a cloud with larger $r_{\text{eff,C}}$ but smaller $\tau$. Therefore, when applying the retrieval all solutions with $r_{\text{eff,C}} < 5\,\mu\text{m}$ were excluded.

## 4.2 Adjustments and uncertainty estimation

The retrieval algorithm was additionally adjusted to Arctic conditions where open water and ice flows may be present in close proximity. Therefore, the algorithms first distinguishes if measurements are obtained over snow-covered sea ice or open

water surfaces. Even in case of overlying clouds, a separation of sea ice and open water is clearly discernable as shown by Schäfer et al. (2015). A surface with high albedo always enhances the upward radiance above a cloud also for optical thick clouds. Therefore, a fixed threshold of cloud reflectivity at a wavelength below roughly $1000\,\text{nm}$ can be applied to distinguish measured cloud reflectivities obtained above snow or white sea ice and open water. Based on the simulations presented above, the spectral reflectivity at $860\,\text{nm}$ wavelength and a threshold of $\gamma(\lambda_0) > 0.65$ was chosen to classify measurements above

snow and open water. This value might change if solar zenith angles are different to the simulations presented here and if clouds with higher optical thickness are considered.

    If $\gamma(\lambda_0)$ is below the threshold an open water surface is assumed. In that case a bi-spectral retrieval following the radiance ratio approach of Werner et al. (2013) was applied. Cloud reflectivities $\gamma(\lambda_2)$ and the reflectivity ratio $R_3 = \gamma(\lambda_3)/\gamma(\lambda_2)$ were used to retrieve $\tau$ and $r_{\text{eff,C}}$. If $\gamma(\lambda_0)$ is above the threshold a snow surface is assumed. In that case the measurements are

converted into the three reflectivity ratios $R_1$, $R_2$, and $R_3$ and interpolated to the 3D grid of simulated values introduced in Sec. 4.1.

    However, as quantified by Schäfer et al. (2015), in the vicinity of sea ice edges 3D radiative effects do significantly influence the reflected radiation. This effect may range up to horizontal distances of a couple of km depending on cloud and surface geometry. Therefore, such measurement sections have been removed from the analysis as retrieved cloud properties might be

biased by up to $90\,\%$ in the close vicinity of sea ice edge.

    Retrieval uncertainties are estimated by considering the uncertainties of each measured reflectivity ratio expressed by its double standard deviation $2\sigma$. The retrieval is operated for varying each ratio, $R_1$, $R_2$, and $R_3$ separately by adding and subtracting $2\sigma$. This procedure ends up in six solutions for the tri-spectral retrieval over snow surfaces and four solutions for



the bi-spectral retrieval over open water. The median of these solutions was used as retrieval result of $\tau$, $r_{\mathrm{eff,C}}$, and $r_{\mathrm{eff,S}}$, while the standard deviation of all solutions quantifies the retrieval uncertainty $\Delta\tau$, $\Delta r_{\mathrm{eff,C}}$, and $\Delta r_{\mathrm{eff,S}}$.

Independent of these uncertainties caused by the measurements itself, systematic uncertainties due to the assumptions in the forward simulations have to be considered. First, the retrieval algorithm is limited to liquid water clouds only and, therefore, may suffer from an incorrect assumption of the cloud phase. In the Arctic, mixed-phase clouds that are dominated by a liquid water layer at cloud top are frequent (Mioche et al., 2015). In that case, the retrieval algorithm may fail as the ice crystals in these clouds absorb solar radiation at similar wavelengths as the snow surface does. The absorption by the ice crystals may add to the absorption of the snow surface and bias the retrieval results.

Second, limitations of the snow albedo parametrization by Zege et al. (2011) applied in the forward simulations may introduce biases in the retrieved $r_{\mathrm{eff,S}}$. E.g., the parametrization assumes a fixed snow grain shape quantified by the form-factor $A = 5.8$. Typical values of $A$ range approximately between from 5.1 for fractals to 6.5 for spheres. For a cloud free retrieval of snow grain size, which represents the maximum effect expected for cloudy conditions, this implies uncertainties in the retrieved snow effective grain size up to 25 %. However, as discussed by Zege et al. (2011), although the uncertainty of the snow grain shape may produce uncertainties in the retrieved $r_{\mathrm{eff,S}}$, the reflection characteristics of the snow, in particular the spectral albedo are not affected. Both properties, $A$ and $r_{\mathrm{eff,S}}$ change the snow albedo with similar spectral pattern. This similarity allows to attribute uncertainty of $A$ as uncertainties of $r_{\mathrm{eff,S}}$. Therefore, the retrieved cloud properties are independent of the assumption of $A$. This emphasises, that the retrieved $r_{\mathrm{eff,S}}$ has always to be considered as an effective quantity. It represents the snow grain size that has to be used in the specific albedo parametrisation (fixed form-factor $A$) to provide the snow albedo which is most representative for the measurements. Spatial or temporal differences in the retrieved $r_{\mathrm{eff,S}}$ may result either by a change of the geometric size of the snow grains or by changes of the snow grain shape quantified by the form-factor.

## 5 Application to airborne measurements during VERDI

Airborne spectral solar radiation measurements were collected with the Spectral Modular Airborne Radiation measurement sysTem (SMART-Albedometer) during the airborne research campaign Vertical Distribution of Ice in Arctic Clouds (VERDI). In April/May 2012 in total 16 research flights with the Polar 5 aircraft operated by Alfred-Wegener Institute for Marine and Polar Research (AWI) were performed over the Canadian Beaufort Sea which was partly covered with snow-covered sea ice and partly ice free.

The SMART-Albedometer measured spectral solar radiance reflected in nadir direction (2.1° field of view) and downward spectral irradiance with grating spectrometers covering the wavelength range between 350 nm and 2200 nm (Ehrlich et al., 2008; Wendisch et al., 2001). From both quantities cloud reflectivity $\gamma$ was calculated using Eq. 1. The SMART-Albedometer was calibrated radiometrically, spectrally and geometrically in laboratory. The uncertainties of the measurements mostly originate from the radiometric calibration given by the uncertainty of the applied radiation source (traceable to the standards of the National Institute of Standards and Technology, NIST) and the signal to noise ratio that differs with wavelength due to the sensitivity of the spectrometers.





By calculating the ratios $R_1$, $R_2$, and $R_3$, as used in the retrieval algorithm, calibration uncertainties partly cancel. E.g., a bias in the radiometric calibration will affect cloud reflectivities at different wavelengths to the same degree, but does not change the ratios. Summing up all effects and considering typical measurements above snow, a $2\sigma$ uncertainty of 6 % was estimated for $R_1$, while for $R_2$ and $R_3$, 4 % and 12 % uncertainty were considered. For the retrieval over open water, where $\gamma(\lambda_2)$ and

the reflectivity ratio $R_3$ are used, the darker surface (lower signal and lower signal to noise ratio) lead to uncertainties in the observations of 9 % for $\gamma(\lambda_2)$ and 13 % for $R_3$.

Two different 30 min extracts from the observations on 29 April (Case I) and 17 May 2012 (Case II) were selected to test the retrieval algorithm. In both cases a wide area close to the coast line was covered by stratiform boundary layer clouds. While for Case I the cloud top, defined by the boundary layer inversion, reached altitudes of up to 700 m, persistent subsidence driven

by anticyclonic conditions lead to a low cloud top of 200 m in Case II. During the remote sensing observations of these clouds, the Polar 5 aircraft flew in an altitude of about 10,000 ft. The flight tracks and the sections of the flight selected for detailed analysis of the retrieval algorithms are included in Figures 8 and 10.

For the application of the retrieval to real measurements it has to be considered, that a pure snow surface is assumed in the forward simulations. Although a snow thickness of 2 cm will be sufficient to neglect the variation of snow albedo with

snow thickness (Warren, 2013), this constrain might not be valid when observations over sea ice with leads or melt ponds are analyzed. The research flights of VERDI have been performed almost exclusively over the partly sea-ice-covered Beaufort Sea. For the two cases analyzed in detail, observations have been selected where the surface conditions are close to the required pure snow surface. However, potential effects by leads, melt ponds or snow free sea ice are discussed for the individual cases.

### 5.1 Case I - 29 April 2012

On 29 April 2012 the observations were collected exclusively over snow-covered sea ice. The selected measurements were obtained between 16:54-17:21 UTC with solar zenith angle close to $63°$ as assumed in the forward simulations of the retrieval algorithms presented in Section 4.1.

The retrieved cloud and snow properties are presented in Fig. 6. Cloud optical thickness ranged between $\tau = 6$ at the beginning of the presented time series and $\tau = 15$ in the second part. $r_{\text{eff,C}}$ also shows a tendency of increasing values but changed

only slightly in time between 6 μm and 9 μm, while $r_{\text{eff,S}}$ remained almost constant at values around 100 μm. The retrieval uncertainties are indicated by the shaded areas in Fig. 6. In the first part of the time series, up to 17:07 UTC, the retrieval uncertainties are lowest for $\tau$ with about $\Delta\tau \pm 1$, and range at $\Delta r_{\text{eff,C}} \pm 1.5$ μm for $r_{\text{eff,C}}$, and $\Delta r_{\text{eff,S}} \pm 60$ μm for $r_{\text{eff,S}}$. In the second part of the time series, the retrieval uncertainty $\Delta\tau$ significantly increase. This correlates with the increase of cloud optical thickness. Similarly, $\Delta r_{\text{eff,S}}$ shows a slight negative correlation with higher uncertainties for low $\tau$. This is also reflected

by the retrieval grids presented in Fig. 5. With increasing $\tau$ the grid spacing becomes more narrow.

For all measurements, the dependencies of $\Delta\tau$ and $\Delta r_{\text{eff,S}}$ as a function of the retrieved $\tau$ and $r_{\text{eff,S}}$ are shown in Figure 7. For $\Delta\tau$, values of $r_{\text{eff,S}}$ are color-coded in each data point, while for $\Delta r_{\text{eff,S}}$ colors indicate $r_{\text{eff,S}}$. Positive correlations are found for both parameters; the larger the retrieved $\tau$ or $r_{\text{eff,S}}$, the larger their uncertainties. For $\Delta\tau$ uncertainties are larger for small $r_{\text{eff,S}}$ (color code in Figure 7a). In this case, small $r_{\text{eff,S}}$ increase the snow albedo and lower the contrast of clouds





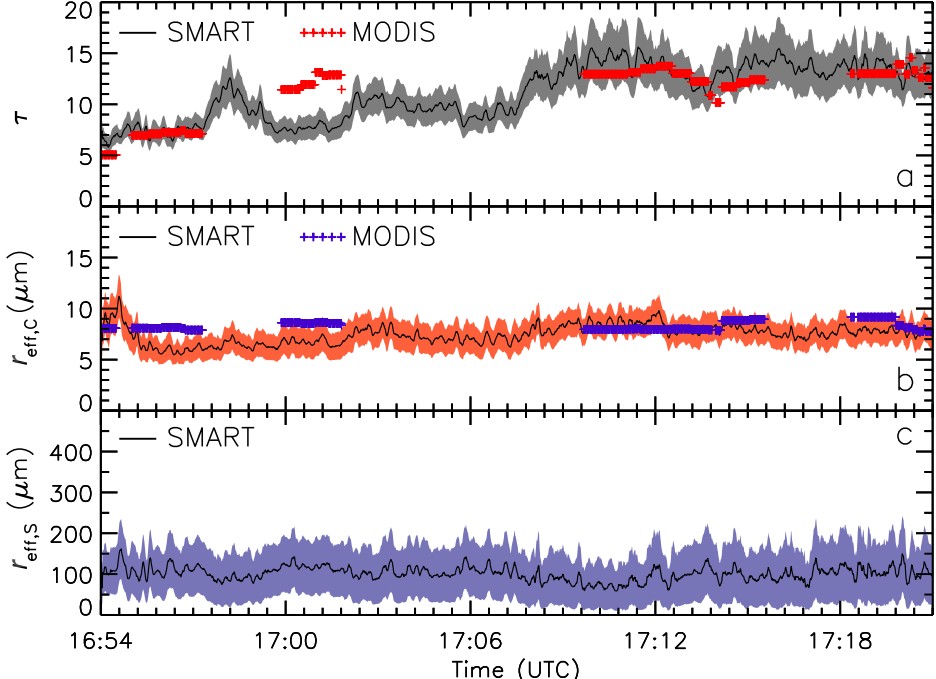

**Figure 6.** Time series of cloud optical thickness $\tau$ (a), cloud droplet effective radius $r_{\mathrm{eff,C}}$ (b) and snow effective grain size $r_{\mathrm{eff,S}}$ (c) retrieved by SMART for Case I on 29 April 2012. Uncertainties of the retrieved properties are indicated by dark shaded areas. When available, results of the MODIS cloud product are given for $\tau$ and $r_{\mathrm{eff,C}}$ (5x5 pixel average).

also at large wavelengths such as $\lambda = 1650\,\mathrm{nm}$ used for the retrieval of $\tau$. Similarly, $\Delta r_{\mathrm{eff,S}}$ depends on the retrieved $\tau$ (color code in Figure 7b); with higher uncertainties observed for large $\tau$. For high optical thickness the clouds begin to mask the surface and information of the surface is lost in the reflected radiation measured above cloud top. Therefore, measurement uncertainties result in higher uncertainties of $r_{\mathrm{eff,S}}$. For $\Delta r_{\mathrm{eff,C}}$, similar dependencies are obtained but with less variability

5    between $\pm 1.2 - 1.7\,\mathrm{\mu m}$ (not shown here). $\Delta r_{\mathrm{eff,C}}$ was found to slightly increase with decreasing $r_{\mathrm{eff,S}}$ and increasing $r_{\mathrm{eff,C}}$.

The snow-covered sea ice below the clouds might have some open or only recently frozen leads which were identified when Polar 5 did fly below clouds after the remote sensing flight leg. From automatic photographs taken on board of Polar 5, the amount of leads was estimated to be lower than 5 % which might explain some of the higher values observed in the retrieved time series of $r_{\mathrm{eff,S}}$.

10    The retrieved cloud and snow properties were compared to satellite observations by MODIS (Wiebe et al., 2013). Figure 8 shows maps of cloud optical thickness $\tau$ (a), droplet effective radius $r_{\mathrm{eff,C}}$ (b) and snow effective grain size $r_{\mathrm{eff,S}}$ (c) retrieved by MODIS. The flight track of Polar 5 is indicated by a black line and overlayed by the retrieval results of SMART. $\tau$ and $r_{\mathrm{eff,C}}$ retrieved by MODIS along the flight track are additionally included in Figure 6. Cloud properties are obtained by the MODIS





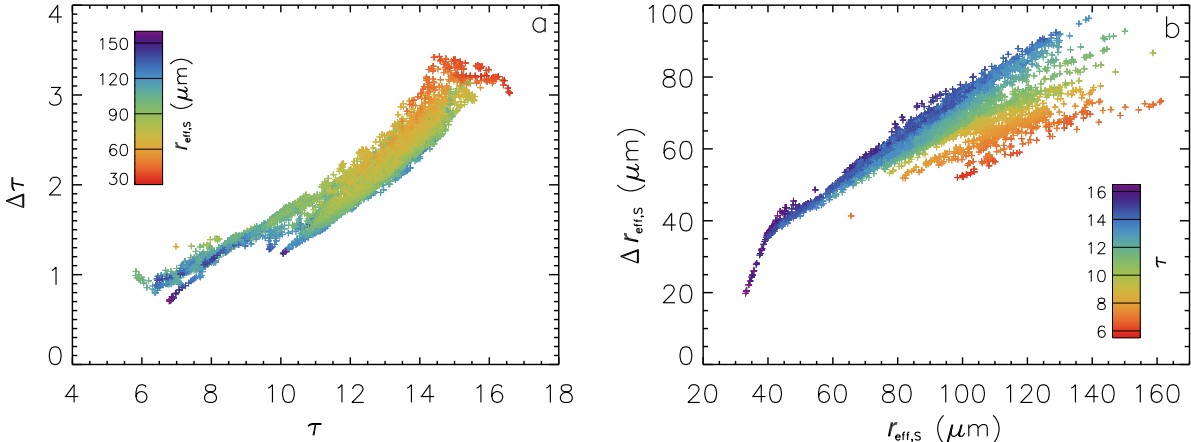

**Figure 7.** Uncertainties $\Delta\tau$ (a) and $\Delta r_{\mathrm{eff,S}}$ (b) as a function of retrieved $\tau$ and $r_{\mathrm{eff,S}}$. The single measurements are color-coded with $r_{\mathrm{eff,S}}$ for (a) and $\tau$ for (b).

cloud product Collection 6 for observations over sea ice using band 6 at $1640\,\mathrm{nm}$ and band 7 at $2130\,\mathrm{nm}$ (Platnick et al., 2017). The snow effective grain size is provided by the Snow Grain Size and Pollution amount (SGSP) retrieval algorithm by Zege et al. (2011). The SGSP is limited to cloud free pixel and, therefore, does not show values below the clouds observed in the same image. For Case I, the AQUA overpass of 20:00 UTC was analyzed. Although, the MODIS data was measured about
5 three hours after the airborne measurements, the stable cloud conditions allow a comparison. Snow grain sizes typically change over longer time scales and, therefore, are directly comparable if no precipitation occurs. The weather station in Tuktoyaktuk close to the coast line did report light precipitation of snow grains but might not be representative for the clouds over the Beaufort Sea. However, a direct comparison of $r_{\mathrm{eff,S}}$ is not possible anyway due to the missing data in cloudy pixel.

The MODIS cloud product in Figure 8 shows $\tau$ and $r_{\mathrm{eff,C}}$ in the same range as retrieved by the airborne measurements. Note
10 that here a longer time series of airborne data is shown as presented in Figure 6. At the edges of the cloud field lower $\tau$ are observed by both MODIS and SMART. For $r_{\mathrm{eff,C}}$ the values retrieved by SMART show the same tendency of lower $r_{\mathrm{eff,C}}$ at the southern cloud edge and increasing $r_{\mathrm{eff,C}}$ towards the western end of the flight track. For large areas of this cloud field the MODIS cloud product did not provide valid solutions what illustrated the limits of current cloud retrieval in Arctic regions.

For $r_{\mathrm{eff,S}}$ no direct comparison is possible. However, the SMART retrieval fills the gap of the cloudy areas not considered
15 in the SGSP retrieval for MODIS. The retrieved $r_{\mathrm{eff,S}}$ are in the same range as observed by MODIS south and north of the cloud field and therefore considered to by consistent with the SGSP product. Lower $r_{\mathrm{eff,S}}$ were detected by MODIS at higher latitudes with some bias due to large leads in the sea ice that are imprinted the retrieval results. At lower latitudes, the snow is strongly influenced by accelerated metamorphism processes by higher temperatures and shows larger $r_{\mathrm{eff,S}}$.





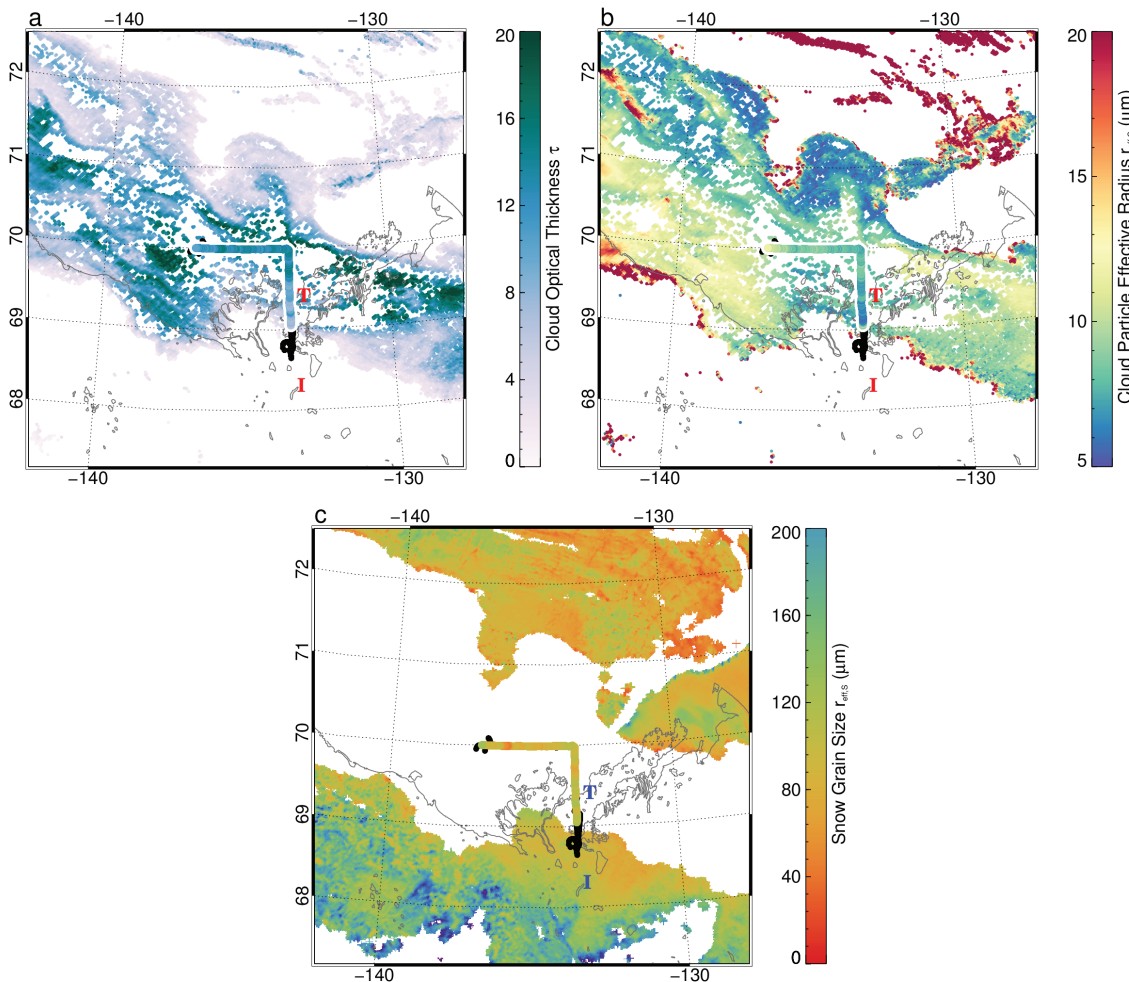

**Figure 8.** Cloud optical thickness $\tau$ (a), cloud droplet effective radius $r_{\mathrm{eff,C}}$ (b) and snow effective grain size $r_{\mathrm{eff,S}}$ (c) retrieved by MODIS and SMART for Case I on 29 April 2012. The total flight track is indicated by a black line and overlayed by the retrieval results of SMART. 'I' and 'T' mark the location of Inuvik and Tuktoyaktuk.

## 5.2 Case II - 17 May 2012

On 17 May 2012 observations were analyzed between 16:45-17:12 UTC on a flight leg crossing a distinct sea ice edge. This transition allows to test the consistency of the proposed retrieval algorithm for observations over snow and open water. Compared to Case I, the cloud altitude was lower with 200 m cloud top altitude indicating a thinner cloud layer. Time of day and, therefore, solar zenith angle were almost identical to Case I and the forward simulations of the retrieval.

5       The retrieved cloud and snow properties for Case II are presented in Fig. 9. Additionally, the light gray shaded time sections indicate measurements above the open ocean while during non-shaded times snow-covered sea ice was present below the





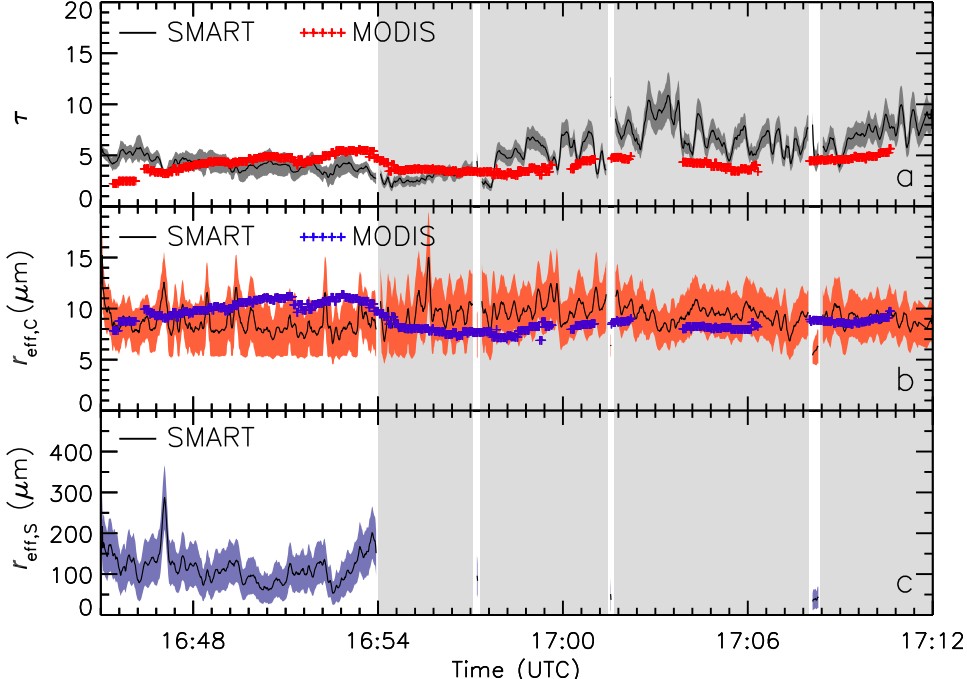

**Figure 9.** Same as Fig. 6 but for 17 May 2012. Light gray shaded times indicate measurements above snow while in non-shades times the ocean was ice free.

clouds. To minimize the impact of 3D-radiative effects at cloud edges, corresponding to Schäfer et al. (2015) and considering the cloud base and top altitude (0-200 m), measurements within a distance of about 400 m to the ice edge were removed from the analysis.

Over sea ice, the retrieved $\tau$ is almost constant at values around 5. Across the sea ice edge, $\tau$ decreases to about 2 and later slightly increases up to $\tau = 10$ with increasing distance to the ice edge. The systematic decrease of $\tau$ over the sea ice edge as retrieved by the airborne measurements extents up to 8 km. As indicated by Schäfer et al. (2015), the radiative field across a straight sea ice edge is affected by the surface albedo transition only up to distances of about 400 m. Therefore, the coincidence of the decrease of $\tau$ with the sea ice edge, observed here, can not be attributed to the retrieval algorithm but rather is natural. Whether, the change of the surface and, therefore, the change of surface latent and sensible heat fluxes did affect the cloud properties across the sea ice edge can not be concluded from this single cross section. As the high resolution MODIS observations indicate, the cloud field did show an oscillation which might have been coincidentally allocated with the sea ice edge at the location of the airborne observations.

The retrieved $r_{\text{eff,C}}$ also varies slightly stronger over open water while over sea ice the retrieved values are almost constant at about $r_{\text{eff,C}} = 8\,\mu\text{m}$ with only short section of higher cloud droplet sizes. Over open water, larger cloud droplets are found



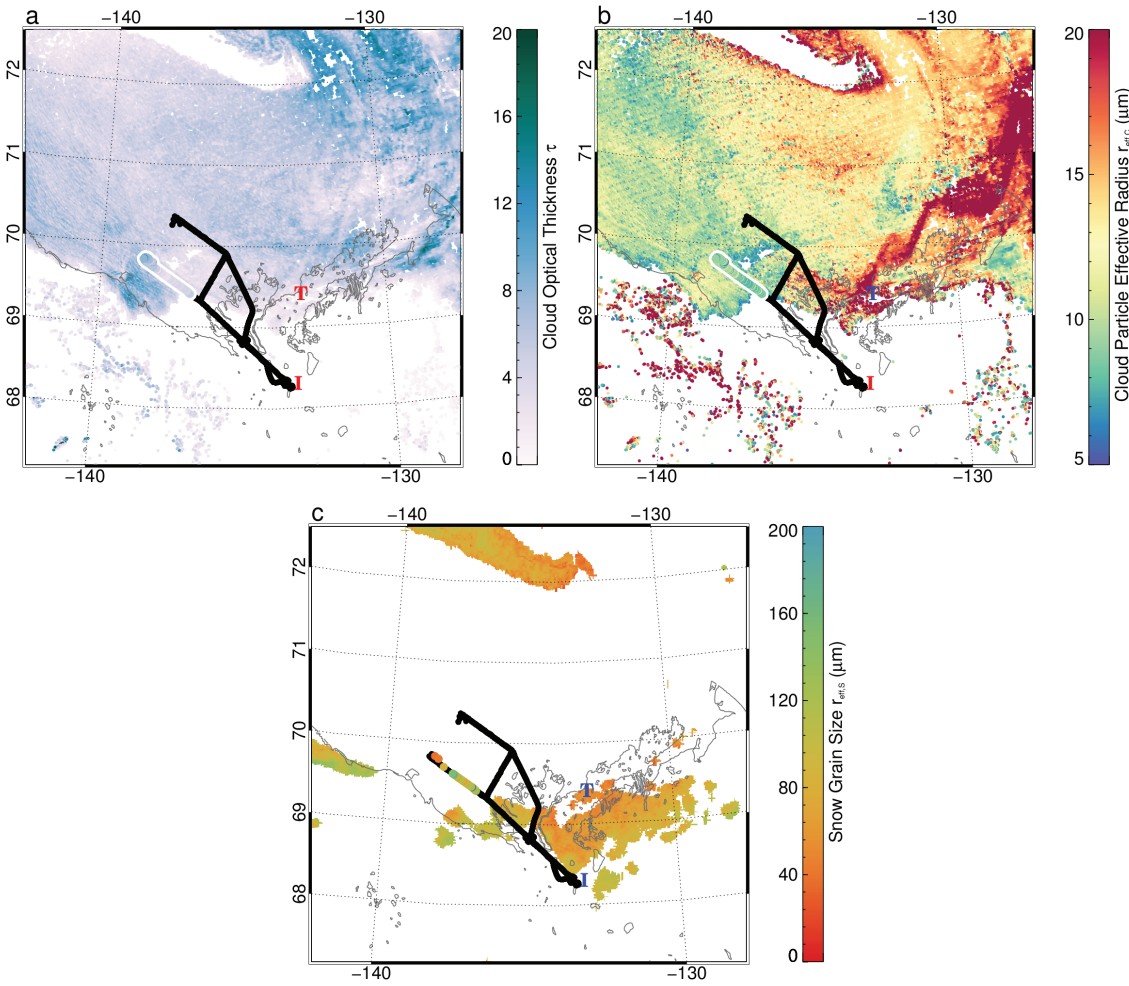

**Figure 10.** Same as Fig. 8 but for 17 May 2012.

with an average of about 10 µm. Close to the sea ice edge, until 17:00 UTC, $r_{\mathrm{eff,C}}$ is found to slightly increase with increasing distance to the sea ice edge simultaneously with the increase of $\tau$. The retrieved $r_{\mathrm{eff,S}}$ shows a slightly higher variability and partly higher values ranging between 50-200 µm compare to Case I. The larger snow grains might result from the different location, the advanced time, and snow metamorphism, or the closer location to the open water. E.g., a systematic decrease

5 of $r_{\mathrm{eff,S}}$ with distance to the sea ice edge is visible (16:52-16:54 UTC, about 6 km distance). In Case II the observations are done above compact fast ice without any leads. Photographs on a flight section in the same area below the clouds showed, that the fast ice was partly not covered by snow what might have cause the higher variability and the single peak with $r_{\mathrm{eff,S}}$ up to 300 µm.




The comparison of cloud and snow properties retrieved by SMART and MODIS is shown in Figure 10 similar to Case I. MODIS results along the flight track are additionally included in Figure 9. For Case II, the MODIS image was observed at 21:25 UTC, more than 4 hours after the airborne observations. Similar to Case I, the temporal variation of the cloud properties is expected to be low as stable dynamic conditions in a high pressure system did prevail during the time of observations and

before. Figure 10 shows that the low values of $\tau$ observed by MODIS were also covered by SMART. The slight increase of $\tau$ in the western end of the flight leg is represented by the retrieval using SMART data. A similar pattern and agreement was found for $r_{\mathrm{eff,C}}$.

The direct comparison of the time series in Figure 9 confirms the general agrement, although differences in the location of cloud fluctuations are obvious. Above sea ice, MODIS observed a steady increase of $\tau$ and $r_{\mathrm{eff,C}}$ and a similar drop at the

ice edge as retrieved by SMART. However, a more quantitative comparison of SMART and MODIS cloud products is not possible due to the time difference between the observations. Differences in the retrieved cloud properties may either result from a wrong assumption of snow albedo or from temporal changes of the cloud. Over open water, the MODIS cloud product provided lower $\tau$ and $r_{\mathrm{eff,C}}$. This is likely caused by the 4 hours difference between both observations. Due to the subsidence in the anticyclonic conditions the cloud top continued to decline and reduced the amount of condensed water, $\tau$, and $r_{\mathrm{eff,C}}$. The

snow effective grain sizes retrieved by SMART are in the range of $r_{\mathrm{eff,S}}$ retrieved by MODIS although the SGSP algorithm could provide results only in small cloud free areas. The single measurements at the western end of the flight leg indicate single ice floes encountered on the ocean and show slightly lower $r_{\mathrm{eff,S}}$. This can be an indication of fresh snow precipitation in this area where cloud optical thickness did increase.

## 6 Conclusions

The retrieval of cloud properties using spectral reflected solar radiation may significantly be biased if the clouds are located over a snow surface or sea ice. An inappropriate assumption of the snow effective grain size results in an incorrect surface albedo at non-visible wavelengths which imprints in the retrieved cloud optical thickness and droplet effective radius. This snow grain size effect is similar to retrieval uncertainties reported by Rolland and Liou (2001) and Fricke et al. (2014) for observations over a variable land surface albedo; only that for snow the surface albedo variability is largest at wavelengths

above roughly 1000 nm, while land surface albedo typically varies in wavelengths below 1000 nm.

For a cloud retrieval using similar wavelengths bands, 1600 nm and 2100 nm, as the MODIS cloud product Collection 6 applies for observations over snow, the snow grain size effect has been quantified on the basis of radiative transfer simulations. For a typical low-level liquid water cloud ($\tau = 4$, $r_{\mathrm{eff,C}} = 10\,\mu\mathrm{m}$) the retrieved cloud properties would differ by up to 50 % if $r_{\mathrm{eff,S}}$ is assumed to be 200 μm instead of the original snow effective grain size of 50 μm, or vice versa. In general:

– The snow grain size effect is largest for small snow grains because the snow albedo changes stronger in the range of small $r_{\mathrm{eff,S}}$, while for larger $r_{\mathrm{eff,S}}$ a saturation of the absorption of radiation is reached.





- The snow grain size effect on retrieved $\tau$ is almost independent of cloud optical thickness. At short wavelengths , used to retrieve $\tau$ ($\lambda = 1600\,\mathrm{nm}$), the snow albedo is still high and always adds to the total reflected radiation. Clouds can not mask this additional reflection of the surface.

- The snow grain size effect on retrieved $r_{\mathrm{eff,C}}$ is strongest for clouds of low optical thickness. At wavelengths used to retrieve $r_{\mathrm{eff,C}}$ ($\lambda = 2100\,\mathrm{nm}$) the snow albedo is close to zero. Therefore, in case of optically thick clouds, the radiation scattered by the clouds dominates the radiation and can mask the additional weak reflection of the surface.

To overcome the snow grain size effect, a method is presented that accounts for changes of the snow grain size in the retrieval algorithm for liquid water clouds by retrieving $r_{\mathrm{eff,S}}$ simultaneously to the cloud properties. A sensitivity study showed that the spectral signatures of cloud and snow properties ($\tau$, $r_{\mathrm{eff,C}}$, $r_{\mathrm{eff,S}}$) significantly differ at specific wavelengths. Three spectral ranges were identified to be most sensitive to the three cloud and snow parameters. At wavelengths between $930 - 1350\,\mathrm{nm}$ the spectral cloud reflectivity is dominated by $r_{\mathrm{eff,S}}$, at $1500 - 1800\,\mathrm{nm}$ by $\tau$, and at $2000 - 2300\,\mathrm{nm}$ by $r_{\mathrm{eff,C}}$.

Based on these spectral sensitivities a retrieval algorithm was designed using reflectivity measurements at $\lambda_1 = 1040\,\mathrm{nm}$ mostly related to $r_{\mathrm{eff,S}}$, $\lambda_2 = 1650\,\mathrm{nm}$ related to $\tau$, and $\lambda_3 = 2100\,\mathrm{nm}$ related to $r_{\mathrm{eff,C}}$. By implementing normalizations in terms of the spectral reflectivity ratios $R_1$, $R_2$, and $R_3$, the impact of measurement uncertainties could be reduced.

The retrieval algorithm was tested for airborne observations by the SMART-Albedometer during VERDI in 2012. Two flight legs, one with closed sea ice and a second flown across a sea ice edge were analyzed. The results and an uncertainty analysis suggest the following conclusions:

- By considering $r_{\mathrm{eff,S}}$, retrieved $\tau$ and $r_{\mathrm{eff,C}}$ are consistent across a sea ice edge where the surface albedo switches from snow-covered sea ice to open water.

- Retrieval uncertainties depend on $\tau$. The thicker the clouds are the stronger they will mask the surface. Less radiation is transmitted into the sub–cloud layer and can be reflected by the surface. This reduces the sensitivity and increases the uncertainties for $r_{\mathrm{eff,S}}$.

- Retrieval uncertainties depend on $r_{\mathrm{eff,S}}$. Small $r_{\mathrm{eff,S}}$ increase the snow albedo and reduce the contract between clouds and snow surface at $\lambda > 1000\,\mathrm{nm}$ increasing the uncertainties of $\tau$ and $r_{\mathrm{eff,C}}$.

- Agreement with MODIS cloud and snow products within the limits of time differences between airborne and satellite observations was found. Differences in the retrieved cloud properties may either result from a wrong assumption of snow albedo or the time difference.

By retrieving cloud properties continuously also along transitions from sea ice to open water, the retrieval algorithm will allow to analyze the impact of surface changes on the cloud microphysical and optical properties. However, retrieval results close to such ice edges or in heterogeneous sea ice conditions are influenced by 3D radiative effects (Schäfer et al., 2015). For the two cases presented here, cloud base altitude and therefore also 3D radiative effects were reduced. Only a limited part of the results had to be excluded from the analysis what might differ for clouds located at higher altitudes (Schäfer et al., 2015).



The presented retrieval assumes that the surface albedo can be described by a pure snow layer of sufficient depth with no influence of the sub-snow surface. However, polar sea ice is not always covered by pure snow. Over new sea ice the snow layer might be still thin and causes the sub-snow surface to reduces the albedo (Malinka et al., 2016; Warren, 2013). In the melting season, melt ponds can change the surface albedo (Grenfell and Perovich, 2004). Locally, melt ponds almost totally

absorb solar radiation at wavelengths larger $800\,\mathrm{nm}$ depending on the pond depth (Lu et al., 2016). However, on larger spatial scales, the albedo of melt ponds and snow areas mix to an albedo with spectral features similar to snow of large grains sizes (Istomina et al., 2015). For such cases, it has to be tested if the proposed retrieval algorithms still can improve the estimated cloud properties. However, the spectral signature of bare sea ice and melt-pond-covered sea is still close to the spectral albedo of pure snow for the wavelengths used in the retrieval. In that case, the retrieved $r_{\mathrm{eff,S}}$ can be interpreted as an effective snow

grain size representing an arbitrary surface albedo (bare sea ice or melt ponds) with the same spectral characteristics as a snow surface with $r_{\mathrm{eff,S}}$.

In this study only liquid water dominated clouds have been analyzed. However, a significant fraction of Arctic clouds are either mixed-phase or ice clouds Mioche et al. (2015). In that case, the retrieval algorithm presented here may fail. The ice crystals in these clouds absorb solar radiation at similar wavelength as the snow surface does. Therefore, the information of

cloud and surface contribution to the reflected radiation might not anymore be sufficiently separated by the wavelengths applied here. Further sensitivity studies have to be performed to identify a different set of wavelengths that is more appropriate for the remote sensing of ice and mixed-phase clouds.

*Acknowledgements.*  We gratefully acknowledge the support by the SFB/TR 172 "ArctiC Amplification: Climate Relevant Atmospheric and SurfaCe Processes, and Feedback Mechanisms (AC)[3]" funded by the DFG. We are grateful to the Alfred Wegener Institute Helmholtz Centre

for Polar and Marine Research, Bremerhaven, Germany for supporting the VERDI campaign with the aircraft and manpower. In addition we like to thank Kenn Borek Air Ltd., Calgary, Canada for the great pilots who made the complicated measurements possible. For excellent ground support with offices and accommodations during the campaign we are grateful to the Aurora Research Institute, Inuvik, Canada.



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
