# Peer review of "Combined retrieval of Arctic liquid water cloud and surface snow properties using airborne spectral solar remote sensing"

_Atmospheric Measurement Techniques, 2017_

## Referee Comment (RC1) · Anonymous Referee #1 · 22 Apr 2017

Summary:

This paper presents a new retrieval method to retrieved concurrently arctic cloud properties (optical depth and effective radius) and surface snow grain sizes. This concurrent retrieval of cloud and surface properties is a distinct improvement in currently used methods. It identifies 3 ratios of reflectances at differing wavelengths that are sensitive to both cloud and surface properties. Its first application of the retrieval method to cases from airborne measurements of spectral albedo in the arctic is presented here.

Overall the paper is very well written with concise description of the retrieval method and first results, and presents a significant contribution to advancing remote sensing of clouds and surface properties in the Arctic. It is recommended for publication with

minor revisions. Besides some specific comments, this paper may be applicable to a broader audience by commenting on the methods' dependence on solar zenith angle.

General Comments:

The paper is well written and well structured. Some minor points to be addressed would make the paper have a slightly more generalized application.

1. The authors used the solar zenith angle of 63° to describe the dependence of the retrieval of cloud optical properties on snow surface albedo grain size. A note on how it would change with a different solar zenith angle would be useful for completeness. His note would also be useful in section 4.2.

2. Section 2.2 is slightly difficult to follow, please refine descriptions.

3. The use of percentages to denote uncertainty is slightly ambiguous and should be better defined.

4. The use of the mean standard deviation per respect to a variable seems to be quite novel. Maybe more description is needed, especially to address the possible covariability of some parameters (i.e. sigma_tau is most variable when there is a high reff_c).

5. Description of the identification process of when the retrieval of snow grain size and cloud property fails would be a useful addition to this paper.

6. The retrieval is applied to data over land although no mention of that in the description of the retrieval methodology description.

7. A note on the availability of surface or in situ measurements for the 2 cases would be helpful.

8. The conclusion is well written, especially with the inclusion of the bullet points.

Specific and Technical Comments:

9. P.2 line 33, exact meaning of sentence not clear, please define what is an improvement of uncertainty by 20%, is it an uncertainty range that is 20% less, or that is it 20% smaller compared to the retrieved value.

10. Fig. 2 could be made clearer if the optical thickness and cloud particle effective radius were put directly on the figure. At least an indication of the low end of the optical thickness and effective radius would be needed.

11. P.5 line 5, cloud reflectivity is also impacted at wavelengths lower than 1000 nm, the word 'only' is erroneous in this case, maybe use a less strict word.

12. P.6 line 9, Sentence slightly difficult to follow.

13. P.6 line 12 Please elaborate or define more clearly 'retrieval forward simulation'

14. P. 8 line 15, typo, should read 'In cases where liquid water clouds are...'

15. Fig. 4 – consider only showing the absolute value of the PCA spectra, for easier comparison to the mean standard deviation values.

16. Typo P.12, line 10, 'too weak' instead of 'to week', sentence would benefit from being more precise.

17. Typo P.12, line 13 'ice floes' instead of 'ice flows'

18. P.12, line 16, revise sentence for the use of the word 'also'

19. Comment, Section 5, the radiometric uncertainties quoted for the ratios seem large considering the calibration uncertainty partially cancel.

---

## Referee Comment (RC2) · Anonymous Referee #2 · 9 May 2017

This paper introduces a tri-spectral method for the retrieval of liquid cloud optical thickness and droplet effective radius, simultaneously with the effective grain size of an underlying snow surface. This method is original and represents a significant contribution to passive Arctic remote sensing. It carries the potential to be implemented for existing imagers (MODIS, VIIRS) which, for some reason, is not emphasized in the current version.

The paper starts by establishing that cloud retrievals are sensitive to the grain size of the underlying snow, especially for small cloud optical thickness. Such clouds are ubiquitous in the Arctic, making this study highly relevant. Grain size retrievals based on MODIS observations exist, and yet the operational cloud retrieval algorithm employs

a climatology for snow-covered regions in the Arctic that does not vary with season and location.

Once the sensitivity of the three key retrieval parameters (cloud optical thickness, cloud droplet radius, snow grain size) have been mapped to spectral radiances that they are most dependent on, a simple lookup table approach is developed in this manuscript. It largely assumes a quasi-orthogonal retrieval grid in the three dimensions of the retrieval and measurement parameter space (see criticism of this aspect below). Finally, a few cases from airborne measurements are used to demonstrate the algorithm.

As noted above, the approach seems highly significant scientifically, and this alone should warrant publication in a major journal such as AMT.

However, there are two main flaws: 1) The language, structure and grammar diminish the potential impact of the manuscript because it becomes hard to read as a result. In sections 4 and 5, it was obvious that it had not been fully proof-read, and it seemed premature to afford it a full review at this point in time. It is beyond the scope of a science review to highlight such issues, but a few examples are listed below. It is in the interest of the authors to revise the language. In some sections (4 and 5 in particular), it could be shortened without losing its content.

2) In general, the science seems sound. However, it is surprising that the retrieval characterization is done without invoking principles of general inverse theory. This is especially important because the retrieval grid is not orthogonal for the most part. This means that there is no 1:1 mapping from observations to retrieval parameters, as the authors clearly acknowledge. But why, then, is the error characterization and propagation done in a fairly "brute force" way as visualized in Figure 7? In the framework of optimal estimation, one could have arrived at a statistically defensible retrieval characterization on the basis of the a-posteriori co-variance while fully taking into account measurement and model uncertainties. That said, a less rigorous error analysis such as done here is acceptable for initial and exploratory studies, as long as it is categorized

as such.

It is hard to tell whether the two above concerns can be alleviated through minor or major revisions; probably the former for the language, and the latter for the scientific approach. If the study were more clearly categorized as exploratory in nature in the revised version (to be followed by a more rigorous paper with a more formal approach routed in inverse theory), the whole manuscript could probably be published with minor changes and a professional copy-editing service.

Other comments:

* It should be mentioned somewhere in the manuscript that this study is strictly valid only for snow-covered surfaces with sufficient geometric (and therefore optical) thickness of the snow. The reference to Malinka (2016) is a bit mis-leading because it sounds as though white ice could be still be represented as snow. This is in stark contrast to multiple publications by, e.g., Perovich for such cases. They show a distinct spectral dependence in the visible wavelength range, and albedos well below 1. Furthermore, "white" ice is not explained. What other ice types are there that might be relevant for cloud remote sensing? A wider literature overview may be helpful.

* p5, L5-11. The reflectance at 1600 nm and 2100 both depend on optical thickness and effective radius; it is simply wrong to decouple them. Figure 2 clearly shows the non-orthogonality of such a lookup table.

* Figure 3a/b are nice visuals of the main direction of this paper; perhaps this could be emphasized more.

* p8: The "standard deviation" and the "PCA" method are insufficiently explain. What is the data set that these methods operate on? Also, the PCA components don't necessarily have to map to a physical parameter as the manuscript seems to suggest.

* p10, L10: Using 860 nm as a reference wavelength for the first ratio is probably a bad idea unless the paper specifically limits itself to snow (rather than including ice). This is

because (as stated above, and described by Perovich) ice has a distinct spectral shape and albedo magnitude at wavelengths below 1000 nm.

* p10, Table 1: This table is reminiscent of a covariance matrix. Why were these relationships not exploited in the framework of optimal estimation? What is the inverse theory foundation of this work?

Grammar/English:

Only a few examples are listed here. In some cases, a spell check could have identified these. In many cases, I did not include line numbers to make it clear that these are not isolated instances but represent a larger problem.

* Inadequate use of emphasis "do cause much lower errors"; "do significantly influence", "field did affect", "did show an oscillation"; "where cloud optical thickness did increase"

* use of "eminent" (p1,L21)

* rep of "retrieve" in short succession in abstract

* consistent punctuation errors (e.g. "analysis showed, that..."; "Both, snow and clouds considerably..."; "...is only possible, if..."

* "Different to land surfaces" ... "In constrast to land surfaces?"

* Therefor [spell checker would pick this up)

* affect = verb; effect = noun (e.g., p3,l2)

* dependence on (not to)

* Section 2.2 (in general); for example: "pretend not to know" [reword]; "range up to"; "Contrarily" [by contrast?];"back through top of" [?]; "procedure ends up with..."

* weight = noun; weigh = verb

* exemplary cloud (this means "outstanding; superb"; surely not the intention)
* p8l15: In case –> When?
* p8l21: "measures" –> "metrics"
* p8L32: "longer wavelength larger" [?]
* spectral pattern[s] result
* "in laboratory"
* "cross almost perpendicular"
* "to week" (too weak?)
* "the measurements itself"
* "Two different extracts from the observations"
* "this constrain might not..." Is it "constraint"?
* "causes...surface to reduces the albedo" [?]

---

## Author Comment (AC1) · 30 Jun 2017

**Reply on Anonymous Referee #1 (AMT-2017-50)**

André Ehrlich[1], Eike Bierwirth[1,2], Larysa Istomina[3], and Manfred Wendisch[1]

[1]Leipzig Institute for Meteorology (LIM), University of Leipzig, Leipzig, Germany
[2]now at: PIER-ELECTRONIC GmbH, Nassaustr. 33-35, 65719 Hofheim-Wallau, Germany
[3]Institute of Environmental Physics, University of Bremen, Bremen, Germany

*Correspondence to:* A. Ehrlich
(a.ehrlich@uni-leipzig.de)

**1 Introduction**

The comments of the reviewer have been helpful to improve the manuscript. Especially the question on the dependence of solar zenith angle on the retrieval results and the missing comparison with in situ measurements provided more understanding of the retrieval algorithm and of the radiative effects between clouds and surface.

The detailed replies on the reviewers comments are given below.

The reviewers comments are given bold while our replies are written in regular roman letters. Citations from the revised manuscript are given as indented and italic text.

**Detailed Replies**

**1. The authors used the solar zenith angle of 63° to describe the dependence of the retrieval of cloud optical properties on snow surface albedo grain size. A note on how it would change with a different solar zenith angle would be useful for completeness. His note would also be useful in section 4.2.**

The solar zenith angle of 63° was used because of the two case studies presented in the manuscript. We now did rerun the sensitivity study for solar zenith angles of 45° and 80°. It was found that the uncertainties in retrieved $\tau$ due to a wrong assumption of the grain size do not significantly change with solar zenith angle. However, for low Sun, $\theta = 80°$, the bias in retrieved $r_{\mathrm{eff,C}}$ was slightly reduced, while for $\theta = 45°$, the uncertainties of the retrieved $r_{\mathrm{eff,C}}$ are increased (see Figure 1). This can be explained by the probability that photons interact with the surface. This is lower for high solar zenith angle. We added this general dependency on solar zenith angle to the revised manuscript:

> *The numbers presented here, were obtained for a solar zenith angle of $\theta_0 = 63°$. For simulations with different solar zenith angles, similar grain size effects are observed. In general, the magnitude of the grain size effect of $\tau$ does not significantly change with $\theta_0$. However, for low Sun, large $\theta_0$, the grain size effect on the retrieved $r_{\mathrm{eff,C}}$ was slightly reduced, while for higher Sun, small $\theta_0$, the effects increase. This is caused by the increased probability that radiation interacts with the surface in case of decreasing solar zenith angle.*

*The snow grain size effect on retrieved $\tau$ does not depend on the solar zenith angle, while the effect on $r_{\mathrm{eff,C}}$ is larger for a higher Sun.*

[Figure]

**Figure 1.** Comparison of retrieval uncertainties for solar zenith angle of $45°$ (left panels) and $80°$ (right panels). Upper plots show the comparison of synthetically retrieved $r_{\mathrm{eff,C}}$ with the original parameter value. Calculations in the upper panels are performed for assuming a larger snow effective grain size of $r_{\mathrm{eff,S}} = 200\,\mu\mathrm{m}$ instead of the original $r_{\mathrm{eff,S}} = 50\,\mu\mathrm{m}$ (crosses) and a smaller snow effective grain size of $r_{\mathrm{eff,S}} = 50\,\mu\mathrm{m}$ instead of the original $r_{\mathrm{eff,S}} = 200\,\mu\mathrm{m}$ (asterisks). In the lower panels all combinations of assumed and original $r_{\mathrm{eff,S}}$ are analyzed for a specific cloud of $\tau = 4$ and $r_{\mathrm{eff,C}} = 10\,\mu\mathrm{m}$.

Additionally, we used the new simulations to estimate the sensitivities of the spectral reflectivity to the three cloud and snow parameter for different solar zenith angles. Not significant differences were found in the spectral separation of the sensitivities. Figures 3 and 1 show the calculated standard deviations and principle component weighting for solar zenith angle of 45° and 80°. The wavelength dependent sensitivities of the cloud reflectivities did change only minor still allowing a separation of the three parameters by measurements at different wavelengths. Similarly, the retrieval grid of the forward simulations was only shifted to one or another direction. The orthogonality and potential ambiguities are almost unchanged. In the revised manuscript we added:

*The results are presented for a solar zenith angle of $\theta_0 = 63°$ but are applicable to larger and smaller $\theta_0$.*

*Although the retrieval was applied to cases with a specific solar zenith angle only, radiative transfer simulations showed that the spectral sensitivities used in the retrieval algorithm are similar in case of smaller or larger solar zenith angles.*

**2. Section 2.2 is slightly difficult to follow, please refine descriptions.**

We reordered some passages in order to improve the readability. See marked up manuscript for all changes.

**3. The use of percentages to denote uncertainty is slightly ambiguous and should be better defined.**

Yes, we agree, percentages can be difficult to interpret. Therefore, we added how the percentage have to be read. In addition, the correct values are always given when percentages are used.

*... the snow grain size effect, expressed by the percentage deviation of the retrieved from the original true value, ...*

**4. The use of the mean standard deviation per respect to a variable seems to be quite novel. Maybe more description is needed, especially to address the possible covariability of some parameters (i.e. sigma_tau is most variable when there is a high reff_c).**

The mean standard deviation was used to provide an efficient way to identify wavelengths that are sensitive to a single cloud or snow parameter. Therefore, we aimed to include all simulations. E.g., for each cloud, a standard deviation of all simulations with different $r_{\mathrm{eff,S}}$ was calculated. $\sigma_{r_{\mathrm{eff,S}}}$ is then derived by averaging these standard deviations for all different clouds. It is right, that for a selection of the set of simulated clouds or snow grain sizes different numbers might be obtained. With the intention to provide a simple retrieval algorithm based on measurements at three wavelength, we did not extended this analysis into more detail. In addition, we finally selected wavelengths corresponding to MODIS bands. A limitation to only a few spectral band will limit the results of a more detailed study investigating the spectral sensitivities for different sub-samples of cloud and snow parameters. To point the use of the mean standard deviations more clearly we added the following sentences in the manuscript.

*E.g., for each cloud, a standard deviation of all simulations with different $r_{\mathrm{eff,S}}$ was calculated. $\sigma_{r_{\mathrm{eff,S}}}$ is then derived by averaging these standard deviations for all different clouds.*

*Similarly, the use of sub samples of the full cloud and snow parameter range investigated here might change the derived values.*

**5. Description of the identification process of when the retrieval of snow grain size and cloud property fails would be a useful addition to this paper.**

In the manuscript, we mentioned, that the retrieval may fail if mixed-phase clouds are present. In that case "fail" does not mean, that the algorithm may stop at some step and does not provide a results. "Fail" refers to wrong results which may be far off the real snow or cloud properties. For mixed-phase clouds, also the cloud ice crystals will absorb the solar radiation at similar wavelengths as snow; the spectral signatures of cloud and snow properties might be merged stronger than for liquid clouds. Therefore, the retrieved snow grain size might be biased by the cloud ice ($r_{\text{eff,C}}$), and vice versa. Still the retrieval would provide a results but this might be wrong. To clarify this, we changed the manuscript to:

*In this case, the retrieval may provide unrealistic cloud properties as the ice crystals absorb solar radiation at similar wavelengths as the snow surface does.*

**6. The retrieval is applied to data over land although no mention of that in the description of the retrieval methodology description.**

Yes, for Case I, the retrieval was also applied for snow covered land surface. These data are only presented in the maps of Figure 8. The time series in Figure 6 does not include the full flight track, as mentioned in the manuscript. However, the application of the retrieval to land surfaces of sufficient snow cover is justified, as the surface below a snow layer does not effect the snow albedo, when the snow layer exceeds a thickness of about 10 cm or more. In the revised manuscript we added a note, that the retrieval was also applied to snow covered land surfaces.

*Note that here a longer time series is shown than presented in Figure 6. This includes areas with snow covered land surfaces, for which the retrieval can be applied assuming that the snow layer is sufficiently thick and the snow albedo is not affected by the underlying surface (Warren, 2013)*

**7. A note on the availability of surface or in situ measurements for the 2 cases would be helpful.**

Measurements of snow grain size on the sea ice have not been conducted during the campaign. Therefore, no reference is available. However, airborne cloud microphysical measurements have been obtained during the flight. Having only one aircraft, Polar 5, the in situ and remote sensing measurements had to be performed subsequently. For both investigated examples, Case I and II, the remote sensing flight legs were flown first. About one hour later the in situ measurements were obtained on the same flight leg. These measurements were included in the revised manuscript and compared to the retrieval results. The following sections have been added:

*Cloud microphysical in situ measurements on board of Polar 5 were use to validate the retrieved $r_{\text{eff,C}}$d. A Cloud Droplet Probe (CDP) provided size resolved cloud particle concentrations in the size range from 2.5 $\mu$m to 46 $\mu$m and*

*corresponding* $r_{\text{eff,C}}$ *(Klingebiel et al, 2015). Using only one aircraft, the in situ and remote sensing measurements had been performed subsequently. For both investigated Cases I and II, the remote sensing flight legs were flown first. Roughly one hour later the in situ measurements were obtained at the same location following the flight track of the remote sensing sequence. Due to the stable meteorological conditions, changes of the cloud properties with time are expected to be small which allows a comparison of in situ and remote sensing data. A reference to validate the retrieved snow grain size is not available because no ground-based measurements on the sea ice have been conducted during VERDI.*

*In situ cloud microphysical measurements of* $r_{\text{eff,C}}$ *had been obtained along the same flight track about one hour after the remote sensing measurements. At cloud top, two derived vertical profiles show* $r_{\text{eff,C}}$ *between 6 μm and 7.5 μm, which is in the range of the retrieval results.*

*The in situ microphysical measurements cover two cloud profiles along the same flight track, one observed above open ocean and one above sea ice. Both profiles showed no difference with* $r_{\text{eff,C}}$ *of about 9 μm at cloud top, which are higher compared to Case I and in agreement with the retrieval results.*

**8. The conclusion is well written, especially with the inclusion of the bullet points.**

Thanks!

**9. P.2 line 33, exact meaning of sentence not clear, please define what is an improvement of uncertainty by 20%, is it an uncertainty range that is 20% less, or that is it 20% smaller compared to the retrieved value.**

Yes, by using "improve by 20 %" the meaning was not sufficiently clear. Indeed, Rolland and Liou (2001) calculated the percentage reduction the the uncertainties range. The absolute reduction differs for different clouds. Therefore, the relative numbers are given. We rephrased the sentence to:

*Rolland and Liou (2001) showed that the retrieval uncertainties of thin cirrus can be reduced by 20 % for optical thickness and by 45 % for ice crystal effective radius when a reasonable estimate of the surface albedo is applied.*

**10. Fig. 2 could be made clearer if the optical thickness and cloud particle effective radius were put directly on the figure. At least an indication of the low end of the optical thickness and effective radius would be needed.**

We added such labels in the revised figure.

**11. P.5 line 5, cloud reflectivity is also impacted at wavelengths lower than 1000 nm, the word 'only' is erroneous in this case, maybe use a less strict word.**

Yes, "only" is not true. We changed the sentence to:

*The simulations illustrate that* $\tau$ *impacts* $\gamma_\lambda$ *primarily at wavelengths larger than* $1000\,\text{nm}$ *where the snow albedo is lower than 0.8, while lower wavelengths are less sensitive to* $\tau$.

**12. P.6 line 9, Sentence slightly difficult to follow.**

That's true, the sentence was way to long. We split now and rephrased to:

> *For liquid water cloud retrievals obtained over snow surfaces with unknown grain size, the snow grain size effect on uncertainties of retrieval results was quantified. Therefore, synthetic measurements obtained from the retrieval forward simulations as introduced in Section 2.1 are applied. For each synthetic measurement defined by $\tau$, $r_{\mathrm{eff,C}}$, and $r_{\mathrm{eff,S}}$ a set of retrieval assuming different values of effective snow grain sizes were performed.*

**13. P.6 line 12 Please elaborate or define more clearly 'retrieval forward simulation'**

The forward simulations of the retrieval are introduced in Section 2.1. We added this reference here.

> *Therefore, synthetic measurements obtained from the retrieval forward simulations as introduced in Section **??** are applied.*

**14. P. 8 line 15, typo, should read 'In cases where liquid water clouds are...'**

Has been corrected in the revised version.

**15. Fig. 4 – consider only showing the absolute value of the PCA spectra, for easier comparison to the mean standard deviation values.**

Yes, we already thought about that, but concluded that the sign of the PCA weightings should not be neglected as it is a valid information. On the other hand, the magnitude itself is important to compare the different principle components. Therefore, we adjusted the plot and showed only absolute values. Original negative weightings are plotted as dashed lines to indicate the sign. However, this new plot looks to busy and we think the main information is lost due to the different line styles. We therefore, would like to keep the original version, but included the alternative version here in the replies (Figure 2).

[Figure]

**Figure 2.** Mean standard deviations of spectral cloud reflectivity $\sigma_\tau$, $\sigma_{r_{\mathrm{eff,C}}}$, and $\sigma_{r_{\mathrm{eff,S}}}$ with respect to a single cloud or snow parameter $\tau$, $r_{\mathrm{eff,C}}$, and $r_{\mathrm{eff,S}}$ calculated for the sets of radiative transfer simulations (panel a). The absolute values of the first three spectral weights $\Gamma_1$, $\Gamma_2$, and $\Gamma_3$ of a principle component analysis are given in panel b. Dashed lines indicate negative values, solid lines positive values of $\Gamma_1$, $\Gamma_2$, and $\Gamma_3$.

**16. Typo P.12, line 10, 'too weak' instead of 'to week', sentence would benefit from being more precise.**

Has been corrected in the revised version.

**17. Typo P.12, line 13 'ice floes' instead of 'ice flows'**

Has been corrected in the revised version.

**18. P.12, line 16, revise sentence for the use of the word 'also'**

Has been corrected in the revised version to:

> *A surface with a high albedo always enhances the upward radiance above a cloud even in the case of optically thick clouds.*

**19. Comment, Section 5, the radiometric uncertainties quoted for the ratios seem large considering the calibration uncertainty partially cancel.**

[Figure]

**Figure 3.** Same es Figure 2 but for solar zenith angle of $45°$ (left) and $80°$ (right)

We carefully calculated the uncertainties of the ratios by considering the uncertainties of the individual radiance measurements. Apart from the radiometric calibration which cancels out for the ratios, the largest contribution to the high uncertainties results form the signal to noise ratio. The noise and dark signal of the spectrometers differ between the two spectrometer types operated in SMART and also within the spectrometer spectral range. Especially the measurements at 2100 nm wavelength are almost at the end of the spectrometer photo diode array where the sensitivity strongly decreases. Therefore, the signal to noise ratio of $R_3$ is quite large. In addition, the radiance is lower at larger wavelengths and even lower in the absorption bands used for the retrieval. These low radiances reduce the signal to noise ratio compared to shorter wavelengths covered by the same spectrometer. This effect can easily lead to uncertainties ranging above 10%.

---

## Author Comment (AC2)

**Reply on Anonymous Referee #2 (AMT-2017-50)**

André Ehrlich[1], Eike Bierwirth[1,2], Larysa Istomina[3], and Manfred Wendisch[1]

[1]Leipzig Institute for Meteorology (LIM), University of Leipzig, Leipzig, Germany
[2]now at: PIER-ELECTRONIC GmbH, Nassaustr. 33-35, 65719 Hofheim-Wallau, Germany
[3]Institute of Environmental Physics, University of Bremen, Bremen, Germany

*Correspondence to:* A. Ehrlich
(a.ehrlich@uni-leipzig.de)

**1  Introduction**

The comments of the reviewer have been helpful to improve the manuscript. We are especially thankful for the plenty corrections of the text and punctuation!

The detailed replies on the reviewers comments are given below.

The reviewers comments are given bold while our replies are written in regular roman letters. Citations from the revised manuscript are given as indented and italic text.

**Detailed Replies**

**It carries the potential to be implemented for existing imagers (MODIS, VIIRS) which, for some reason, is not emphasized in the current version.**

We agree that the proposed method could be implemented for MODIS or VIIRS. The reason why we did not highlighted this explicitly is that we think we would have to prove it when making this statement. In the revised version we carefully addressed the possibility to use the method for satellite imagers; once in the algorithm description and once in the conclusions.

> *Except for $\lambda_1$, all wavelengths that were chosen for the algorithm are covered by the satellite imagers MODIS and VIIRS. To apply the algorithm to global observations by these instruments, $\lambda_1$ can be exchanged by the 1240 nm wavelength band where cloud reflectivity is still most sensitive to $r_{\mathrm{eff,S}}$.*

> *Therefore, the proposed retrieval method has some potential to be implemented for existing spaceborne imagers such as MODIS or VIIRS. Due to the limited number of spectral bands, for these two instrument $\lambda_2$ would have to be exchanged by the 1240 nm wavelength band where cloud reflectivity is still most sensitive to $r_{\mathrm{eff,S}}$.*

**1) The language, structure and grammar diminish the potential impact of the manuscript because it becomes hard to read as a result. In sections 4 and 5, it was obvious that it had not been fully proof-read, and it seemed premature to afford it a full review at this point in time. It is beyond the scope of a science review to highlight such issues, but a**

**few examples are listed below. It is in the interest of the authors to revise the language. In some sections (4 and 5 in particular), it could be shortened without losing its content.**

We are sorry about the incorrect language and grammar. We revised the manuscript and tried to improve as much as possible. However, as we are no native speakers, we suggest to have a professional copy-editing by the journal. Languages changes in the text can be found in the highlighted manuscript version.

**2) In general, the science seems sound. However, it is surprising that the retrieval characterization is done without invoking principles of general inverse theory. This is especially important because the retrieval grid is not orthogonal for the most part. This means that there is no 1:1 mapping from observations to retrieval parameters, as the authors clearly acknowledge. But why, then, is the error characterization and propagation done in a fairly "brute force" way as visualized in Figure 7? In the framework of optimal estimation, one could have arrived at a statistically defensible retrieval characterization on the basis of the a-posteriori co-variance while fully taking into account measurement and model uncertainties. That said, a less rigorous error analysis such as done here is acceptable for initial and exploratory studies, as long as it is categorized as such.**

Yes, this is right! To fully understand the uncertainties of the retrieval other statistical methods would have been needed. However, the intention of our study was to illustrate the concept of such a tri-spectral retrieval that allows to combine cloud and snow optical properties. Therefore, we kept the uncertainty analysis simple, also because the retrieval was only applied to two selected cases of specific solar zenith angle and to the nadir viewing direction of the airborne measurements. A more detailed analysis should also take into account the different sensitivities for different viewing geometries. We therefore changed the manuscript at different locations in order to point out clearly, that the manuscript presents only a feasibility study to the method but not a fully developed and characterized retrieval algorithm. A detailed comprehensive uncertainty analysis is beyond the scope of this paper and can be part of studies where the method is applied to a more general data set.

*In a feasibility study, spectral cloud reflectivity measurements collected by the Spectral Modular Airborne Radiation measurement sysTem (SMART) during the research campaign Vertical Distribution of Ice in Arctic Mixed-Phase Clouds (VERDI, April/May 2012) were used to test the retrieval procedure.*

*In a feasibility study in Section 5, the algorithm that is limited to cases of liquid water clouds is applied to two specific cases,...*

*The retrieval algorithm was tested in a feasibility study for airborne observations by SMART during VERDI in 2012.*

*For this first application of the new tri-spectral retrieval algorithm, a rather simplistic analysis was applied. A more general understanding of the retrieval sensitivities and uncertainties can be achieved by optimal estimation techniques, which is beyond of the scope of this paper.*

**\* It should be mentioned somewhere in the manuscript that this study is strictly valid only for snow-covered surfaces with sufficient geometric (and therefore optical) thickness of the snow. The reference to Malinka (2016) is a bit mis-**

**leading because it sounds as though white ice could be still be represented as snow. This is in stark contrast to multiple publications by, e.g., Perovich for such cases. They show a distinct spectral dependence in the visible wavelength range, and albedos well below 1.**

We are aware that snow and white ice albedo are different at visible wavelengths. But there are reasons why the retrieval approach is still applicable, with larger uncertainties, to white ice or a mixtures of sea ice, open leads or melt ponds. In this cases still more accurate cloud properties compared to methods assuming a fixed snow/sea ice albedo can be derived, because the tri-spectral retrieval considers the change of the spectral albedo at the three wavelengths.

1) Visible wavelengths where white ice albedo and snow albedo have different spectral signatures are not used in the proposed retrieval. At larger wavelengths, where mostly the absorption by ice and not the scattering processes are relevant, the spectral pattern of white ice albedo, which is not water saturated, i.e. dry white ice, is similar to a snow albedo. Therefore, also the snow albedo model used in our retrieval approach can be applied to approximate the albedo of white ice in these spectral ranges. But of course not in the visible part of the solar spectrum. More sophisticated albedo models such as by Malinka et al. (2016) have to be used to construct the full spectral albedo. But then either measurements at visible wavelengths have to be included or information on the effective optical thickness are needed. In the revised manuscript we corrected:

> *Additionally, at wavelengths larger than 1000* nm *the albedo of white sea ice that is not covered by snow and not water saturated, i.e. dry white ice, is lower than that of snow-covered sea ice and, therefore, can be characterized by larger effective snow grain sizes (Malinka et al., 2016).*

2) The retrieval does not provide a retrieval of the full spectral surface albedo. The aim is to retrieve a parameter, the effective snow grain size $r_{\mathrm{eff,S}}$, which determines the surface albedo in the applied snow albedo model. Of course, we can not reconstruct the albedo of sea ice in the full solar spectral range as the snow albedo model does not capture this. Using three wavelengths, only information at these wavelengths are available. Therefore, we can only say that a certain snow albedo model fits best to the measurements at these wavelengths. The snow albedo model used in the algorithm is based only on one parameter, the effective grain size. In this sense, the effective snow grain size is only a parameter defining the most likely surface albedo at the applied wavelengths. That's why we always name it "effective". Considering the "effective" character of $r_{\mathrm{eff,S}}$, also the albedo of white ice can be parameterized using $r_{\mathrm{eff,S}}$ (see 1).

3.) Other common satellite grain size retrievals that are only applied in clear sky condition, such as mentioned and presented in the study, do consider similar snow albedo models that are based only on the effective grain size $r_{\mathrm{eff,S}}$.

4.) However, to avoid larger uncertainties by white sea ice, the cases presented in the manuscript are carefully selected and are dominated by snow-covered sea ice. We highlighted this in the revised manuscript at different positions. E.g.:

> *..., observations have been selected where the surface conditions are close to the required pure snow surface.*

> *Photographs on a flight section in the same area below the clouds showed that the fast ice was partly free of snow, which may have caused the higher variability and the single peak of* $r_{\mathrm{eff,S}} = 300\,\mu\mathrm{m}$

*However, the spectral signature of white sea ice and melt-pond-covered sea is close to the spectral albedo of pure snow for the wavelengths used in the retrieval. In that case, the retrieved $r_{\text{eff,S}}$ is interpreted as an effective snow grain size representing an arbitrary surface albedo (white sea ice or melt ponds) with the same spectral characteristics above 1000 nm wavelength as a snow surface with $r_{\text{eff,S}}$.*

**Furthermore, "white" ice is not explained. What other ice types are there that might be relevant for cloud remote sensing? A wider literature overview may be helpful.**

Of course also other sea ice types are relevant for cloud remote sensing. Unfortunately, the proposed retrieval algorithm is not capable of considering these, e.g., blue ice, because of the different spectral albedo. Therefore, the algorithm is proposed for snow-covered surfaces. The only exceptions is the dry white ice, because it approximately can be treated similar to snow of large grains at wavelengths larger 1000 nm. We therefore, do not want to include the definition of other ice types because this might be misleading and give the impression that the retrieval might be also working for such surfaces. Which is not the case. For "white ice" a reference is given in the introduction.

**\* p5, L5-11. The reflectance at 1600 nm and 2100 both depend on optical thickness and effective radius; it is simply wrong to decouple them. Figure 2 clearly shows the non-orthogonality of such a lookup table.**

We did not state that both parameters are decoupled. We explicitly state on Page 6, Lines 5-7 that "... the reflectivities at both wavelengths are coupled to both cloud parameters.". Our only idea why the reviewers made this comment is that the sentence Page 5, Lines 9-11 lead to the impression, that the parameters are only linked to one wavelength. This sentence was used to describe the general idea of an bi-spectral cloud retrieval, were the different dependencies are utilized to derive cloud optical thickness and effective radius. To avoid the misinterpretation of this sentence, we rephrased:

*The retrieval uses the different dependencies of $\gamma_{1600\,\text{nm}}$ (less-absorbing wavelength) and $\gamma_{2100\,\text{nm}}$ (high-absorbing wavelength) on cloud optical thickness and cloud droplet effective radius and basically follows the method by **?**.*

**\* Figure 3a/b are nice visuals of the main direction of this paper; perhaps this could be emphasized more.**

To emphasize the major results shown in these plots, we added in the abstract the following conclusions:

*he impact of uncertainties of $r_{\text{eff,S}}$ is largest for small snow grain sizes. While the uncertainties of retrieved $\tau$ are independent of the cloud optical thickness and solar zenith angle, the bias of retrieved $r_{\text{eff,C}}$ increases for optically thin clouds and high Sun.*

**\* p8: The "standard deviation" and the "PCA" method are insufficiently explain. What is the data set that these methods operate on? Also, the PCA components don't necessarily have to map to a physical parameter as the manuscript seems to suggest.**

The data set used for these calculations was given in the first sentence of the section. In the revised manuscript, we more clearly described the use of the mean standard deviations to subsets of the simulations by adding the following sentences in the manuscript.

> *E.g., for each cloud, a standard deviation of all simulations with different $r_{\mathrm{eff,S}}$ was calculated. $\sigma_{r_{\mathrm{eff,S}}}$ is then derived by averaging these standard deviations for all different clouds.*

> *Similarly, the use of sub samples of the full cloud and snow parameter range investigated here might change the derived values.*

For the PCA it was already stated that we "... applied to the full set of simulations". It is true, that the weightings of the PCA do not necessarily have to map a single physical parameter. But obviously for the set of simulations presented in the manuscript this is the case. To avoid the impression, that this separation is given by theory, we changed the following sentence to:

> *Corresponding to the cloud and snow parameters changed in the simulations, the spectral weights $\Gamma_1$, $\Gamma_2$, and $\Gamma_3$ of the first three principle components are found to be associated with $\tau$ ($\Gamma_1$), $r_{\mathrm{eff,C}}$ ($\Gamma_2$), and $r_{\mathrm{eff,S}}$ ($\Gamma_3$).*

**\* p10, L10: Using 860 nm as a reference wavelength for the first ratio is probably a bad idea unless the paper specifically limits itself to snow (rather than including ice). This is because (as stated above, and described by Perovich) ice has a distinct spectral shape and albedo magnitude at wavelengths below 1000 nm.**

Yes, for the current version we would like to limit the retrieval to snow covered surfaces and not extend it to other sea ice types. We first would like to illustrate the main idea that information of the surface albedo is still imprinted in spectral radiance measurements above clouds. Therefore, we prefer using measurements at 860 nm because at this wavelength the differences in snow and sea ice albedo is low. Of course, this similarity is completely gone once the sea ice starts melting or its surface gets water saturated. However, dry white ice with some scattering layer on top (of typically granular crystals) is comparable to the snow with not too fine grains. The publication by Malinka et al. (2016) states this exact point.

Also snow impurities do affect the spectral albedo less at 860 nm compared to shorter wavelengths. It is possible, that our approach can be extended to shorter wavelengths in order to detect a drop in the surface albedo at visible wavelengths related to different ice types. But therefore, we would need to use a more detailed snow/sea ice albedo model which is beyond the scope of this study.

**\* p10, Table 1: This table is reminiscent of a covariance matrix. Why were these relationships not exploited in the framework of optimal estimation? What is the inverse theory foundation of this work?**

For us, the main intention of this study is to present the feasibility of retrieving cloud and snow properties simultaneously by making use of the spectral information that is provided by spectral measurements of cloud reflectivity. Therefore, we did not set the focus of our attention on the inverse theory but applied rather simple methods; spectral standard deviation and principle component analysis to identify sensitive spectral regions; propagation of normally distributed uncertainties through the retrieval to derive retrieved parameters with uncertainty estimate. As the sensitivities seems sound and the retrieval algorithm already provides reasonable results, we did not put more effort in improving the inverse method at this stage. Therefore, we now

pointed out more clearly that the study should be seen as a feasibility study. Later, more detailed analysis of sensitivities and uncertainties may follow. For changes in the manuscript see answer to comment number 2).

**Grammar/English:**

Thanks a lot for identifying all these mistakes. We are sorry, that these were not found by ourself and collecting all increased your work. We corrected all and tried our best to eliminate other incorrect grammar and typos.